# Fixational drift is driven by diffusive dynamics in central neural circuitry

Nadav Ben-Shushan[1,4], Nimrod Shaham[1,2,4], Mati Joshua [3,5] & Yoram Burak [1,3,5 ✉]

During fixation and between saccades, our eyes undergo diffusive random motion called fixational drift. The role of fixational drift in visual coding and inference has been debated in the past few decades, but the mechanisms that underlie this motion remained unknown. In particular, it has been unclear whether fixational drift arises from peripheral sources, or from central sources within the brain. Here we show that fixational drift is correlated with neural activity, and identify its origin in central neural circuitry within the oculomotor system, upstream to the ocular motoneurons (OMNs). We analyzed a large data set of OMN recordings in the rhesus monkey, alongside precise measurements of eye position, and found that most of the variance of fixational eye drifts must arise upstream of the OMNs. The diffusive statistics of the motion points to the oculomotor integrator, a memory circuit responsible for holding the eyes still between saccades, as a likely source of the motion. Theoretical modeling, constrained by the parameters of the primate oculomotor system, supports this hypothesis by accounting for the amplitude as well as the statistics of the motion. Thus, we propose that fixational ocular drift provides a direct observation of diffusive dynamics in a neural circuit responsible for storage of continuous parameter memory in persistent neural activity. The identification of a mechanistic origin for fixational drift is likely to advance the understanding of its role in visual processing and inference.

[1] Racah Institute of Physics, The Hebrew University of Jerusalem, Jerusalem, Israel. [2] Swartz Program in Theoretical Neuroscience, Center for Brain Science, Harvard University, Cambridge, MA, USA. [3] Edmond and Lily Safra Center for Brain Sciences, The Hebrew University of Jerusalem, Jerusalem, Israel. [4]These authors contributed equally: Nadav Ben-Shushan, Nimrod Shaham. [5]These authors jointly supervised this work: Mati Joshua, Yoram Burak. ✉email: yoram.burak@elsc.huji.ac.il

In order to explore the fine details of a visual scene, we fixate our gaze on specific areas of interest[1,2]. Yet even during fixation the eyes are not completely stationary. Over intervals that typically last a few hundred milliseconds, the eyes exhibit continuous motion called fixational drift, flanked by microsaccades[3–5]. Eye trajectories during fixational drift are smooth, but are highly variable and are characterized by the statistics of a super-diffusive random walk[6–9]. The role of this irregular smooth motion in vision has been extensively studied and debated in the past few decades. It was shown that both drifts and microsaccades help prevent perceptual fading during fixation[10,11]. In addition it has been proposed that fixational drift increases the information carried by retinal spikes on the visual stimulus, thereby aiding high acuity vision[4,12–16]. On the other hand, it has been argued that fixational drift poses a computational challenge for high acuity inference in the visual cortex, since attempting to overcome retinal spiking noise by simple temporal averaging would smear out fine visual features[15,17,18].

While extensive effort has been devoted to understand the functional role of fixational drift in vision, the mechanisms responsible for this motion have remained unidentified. It is not even known whether the origin of fixational drift is peripheral – arising, e.g., from noisy dynamics of the ocular muscles[19], or whether fixational drift arises in central brain circuits in similarity to saccades[3,7,20]. Recent behavioral studies indicated that visual feedback mechanisms, which likely involve central brain circuits in the visuomotor pathway, can modulate the statistics of the motion[4,14,21–25]. However, an active visuomotor response to a stimulus is not required to elicit fixational drift, since it is observed even in complete darkness[26]. The stochastic nature of the motion, both in the presence and in the absence of visual stimuli, suggests that it is primarily driven by noise which may arise in various stages along the oculomotor pathway.

So far, direct evidence for the control of fixational eye drifts by neural activity has been lacking[27]. The main difficulty in seeking such evidence arises from the small amplitude of fixational drifts in comparison with other types of eye movement. In human subjects it is highly challenging to measure the minute details of this motion[1], and measurements of single neuron activity are not available. Therefore, we focus from here on non-human primates, whose control of eye movements is highly similar to that of humans both during saccades and pursuit[28], and during fixational eye movements[1].

During saccades and smooth pursuit, eye trajectories can be predicted quite precisely from the spiking activity of single oculomotor neurons (OMNs). Correspondence between eye movements and neural activity has been established also during microsaccades[29]. However, a correspondence between the position of the eye and single OMN activity has not been demonstrated during fixational drift. It is highly challenging to test for such a relationship, because the subtle changes that are expected to occur in the firing rate of individual OMNs during fixational drift are largely masked by their spiking noise.

Here we show that a systematic relationship does exist between fixational eye motion and OMN activity. Furthermore, we show that most of the variability in fixational eye drifts arises upstream of the OMNs. This result establishes that the main drive for fixational drift lies in more central neural circuitry. Next, we point to a likely source of this motion in the oculomotor integrator, a memory circuit in the brainstem which is responsible for maintenance of a steady eye position between saccades. We propose that fixational eye drifts are driven by random diffusion along a line attractor that characterizes the dynamics of the oculomotor integrator[30–32], as predicted by theoretical works that examined how noise influences the maintenance of continuous-parameter working memory in persistent neural activity[33–35]. Using a theoretical model, constrained by the parameters of the primate oculomotor system, we show that this mechanism naturally explains both the magnitude of the motion, and key features of its statistics.

## Results

To test for a systematic relationship between OMN activity and fixational eye drifts, we analyzed a large data set of OMN extracellular recordings in the rhesus monkey. These were collected simultaneously with precise measurements of eye trajectories using a search coil. Recordings were made while two monkeys moved their eyes to track repeated presentations of a target presented on a screen, initially at rest and then moving at constant speed (Fig. 1a). First, we fitted a linear combination of horizontal eye position, velocity and acceleration to predict the firing rate of single OMNs during large eye movements[36] ($n = 57$ cells, Fig. 1b). Fitted parameters were in agreement with previously reported results[36,37].

OMNs exhibit highly regular firing, with a typical coefficient of variation (CV) of the interspike interval distribution of ~0.07[38]. During fixational drift, however, the variability in the activity of a single OMN is still far too large to identify a correspondence with the small changes in firing rate predicted by eye motion over a single trial (Fig. 1b, lower panel). Over the hundreds of trials available from each cell, estimates of the correlation coefficients, during fixation, between the spiking rate of single OMNs and their predictors based on the eye trajectory were noisy (Fig. 1c), but significantly deviated from zero over the population of 57 cells (Fig. 1d), providing us with initial evidence that neural activity, upstream of the muscles, is correlated with fixational drift.

To facilitate subsequent analysis, we replaced the standard approach discussed above, in which the firing rates of OMNs are predicted from the eye trajectory by a complementary analysis, in which the eye trajectory is predicted from the OMN spikes. This allows us below to quantify shared neural variability directly in terms of its contribution to the eye movements, instead of the firing rates. We thus inverted the fit described above (Methods) to obtain for each OMN a double exponential filter (Fig. 2a), whose convolution with the OMN spikes constitutes an unbiased estimator of the eye trajectory. Fitted time constants (Fig. 2b) were in agreement with previous reports[36,39]. As expected due to the spiking noise in the activity of individual OMNs, predicted eye trajectories were highly variable compared to the actual eye motion during fixational drift (Fig. 2c). Accordingly, correlation coefficients between the predicted and actual eye position differences across 350 ms fixational drift segments (denoted by $R$, Fig. 2d) were small. Furthermore, due to the limited number of trials available for each neuron, the estimates of the correlation coefficients themselves were noisy (gray histogram in Fig. 2d). Nevertheless, their average value $\langle R \rangle = 0.17$ (angled brackets represent an average over the OMN population) was statistically different from zero (one sided $t$ test, $p = 3.8 \times 10^{-12}$, Fig. 2d and Extended Data Fig. 1). Overall, our results (Figs. 1 and 2) indicated that fixational drift is correlated with neural activity.

**Central source**. We next sought to identify whether the weak correlations with eye motion, observed in the noisy activity of single OMNs, are indicative of a dominant central contribution to the motion, upstream of the OMNs. First, we examined the squared correlation coefficient $R^2$ (Fig. 2d). This quantity represents the fraction of the motion variance (in 350 ms intervals) that can be linearly predicted based on the OMN spikes, using the estimator described above. Hence, it was of interest to estimate the average of $R^2$ across the neural population, which we denote by $\langle R^2 \rangle$. To do so, we first evaluated the average of $R$ across the

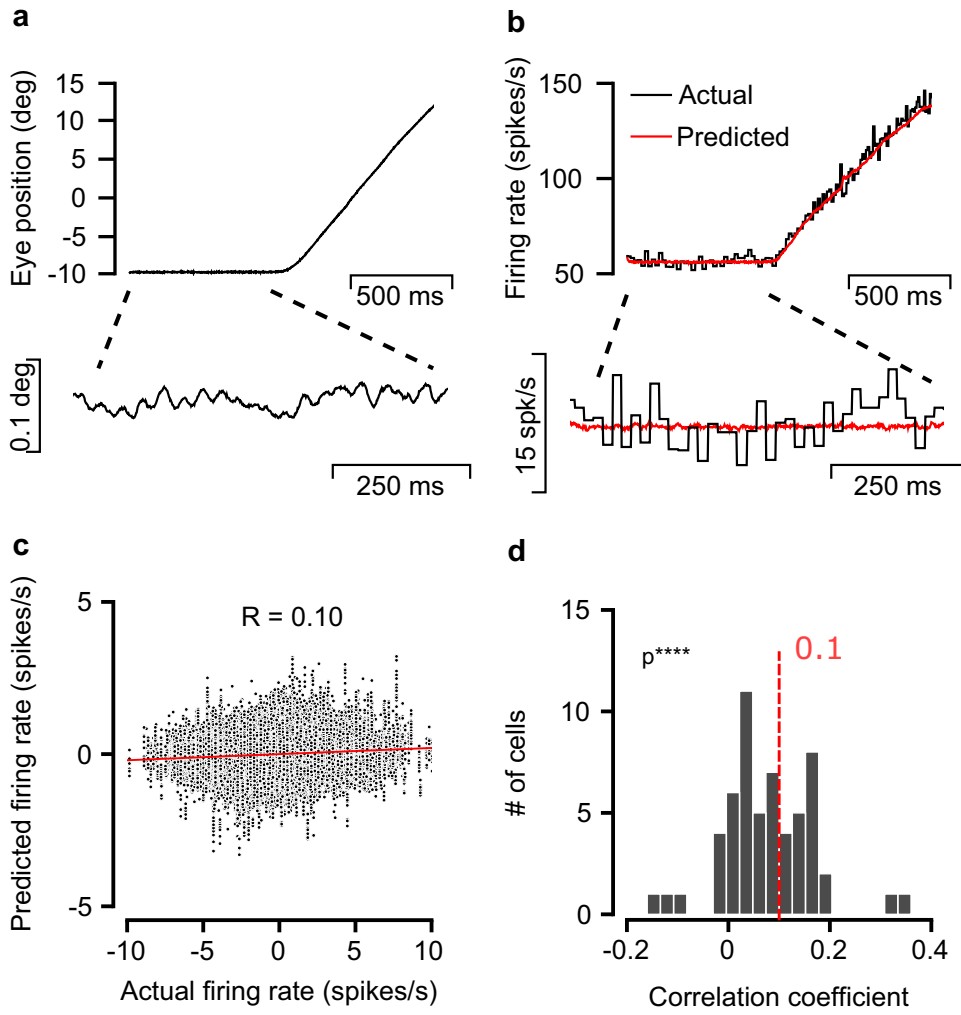

**Fig. 1 Eye variability during fixation is correlated with motoneuron activity. a** Single recorded horizontal eye trajectory consisting of fixation followed by smooth pursuit. Lower panel, fixational period expanded and smoothed with a Savitzky-golay filter of order 3 (21 ms window, data sampled at 1 KHz). **b** Actual and predicted firing rate of a single OMN during the trajectory shown in (**a**). Lower panel, zoom in on fixational segment. Note that the actual firing rate is much more variable than the predicted rate. The predicted firing rate based on the spikes was obtained by fitting the parameters $k, r, m, E_T$ in eq. (2) to minimize a mean squared error loss function, eq. (3) (Methods). **c** Predicted vs. actual firing rate of a single cell during fixation (mean rate subtracted in both axes). Each point represent the firing rate at a single trial at specific time. Red line: result of linear regression analysis, correlation coefficient: R = 0.10. **d** Distribution of correlation coefficients between actual and predicted rate, collected from 57 cells in two monkeys. One sided $t$ test, $p = 3 \times 10^{-9}$.

neural population ($\langle R \rangle = 0.17 \pm 0.02$, mean ± SEM), using the distribution shown in Fig. 2d (individual estimates of $R$ for all neurons are shown in Extended Data Fig. 1a, c). By noting that $\langle R^2 \rangle \geq \langle R \rangle^2$ (since $\mathrm{Var}(R) = \langle R^2 \rangle - \langle R \rangle^2 \geq 0$) it was possible to deduce that $\langle R^2 \rangle \gtrsim 0.029$: on average, at least ~2.9% of the variability in eye motion during a 350 ms interval can be predicted based on the activity of a single OMN (see also SI Notes). For comparison, consider a scenario in which the upstream drive to OMNs is completely static during fixational drift, and the motion is driven by the intrinsic variability of the OMNs, which is independent in the different neurons. Since there are thousands of OMNs in the primate oculomotor system[40,41], a single OMN would be expected to explain ~0.1% of the variability in the eye motion, which is far smaller than the observed explanatory power. We could thus infer that the activity of different OMNs during fixational drift covaries (see also SI Notes). It is unlikely that this shared variability among OMNs is generated intrinsically within the abducens, from which we recorded the neural activity (see Discussion). Hence, this result indicates that the eye motion

is driven, at least in part, by a common upstream input[42] (see also SI Notes).

The above analysis indicates that there is a common input to OMNs, but since typical values of $R^2$ were small compared to unity, it was not clear whether the common input provides a main drive for the motion. Importantly, the values of $R^2$ are small because single OMN estimates of the eye motion are dominated by the irregularity in the spiking of the OMNs, but the variability arising from the spiking noise need not dominate the overall motion. The central input could, in principle, be the dominant drive for the motion because its contribution adds up coherently downstream of the OMNs, in contrast to the contribution of the spiking noise which is independent in different neurons.

To estimate the contribution of the central source to the motion, we assumed that there are three types of sources that linearly contribute to the measured eye motion: first, a central signal that feeds into all the OMNs. Second, noise in the activity of individual OMNs, which is independent in different units (see Discussion). Third, noise downstream of the OMNs, which may

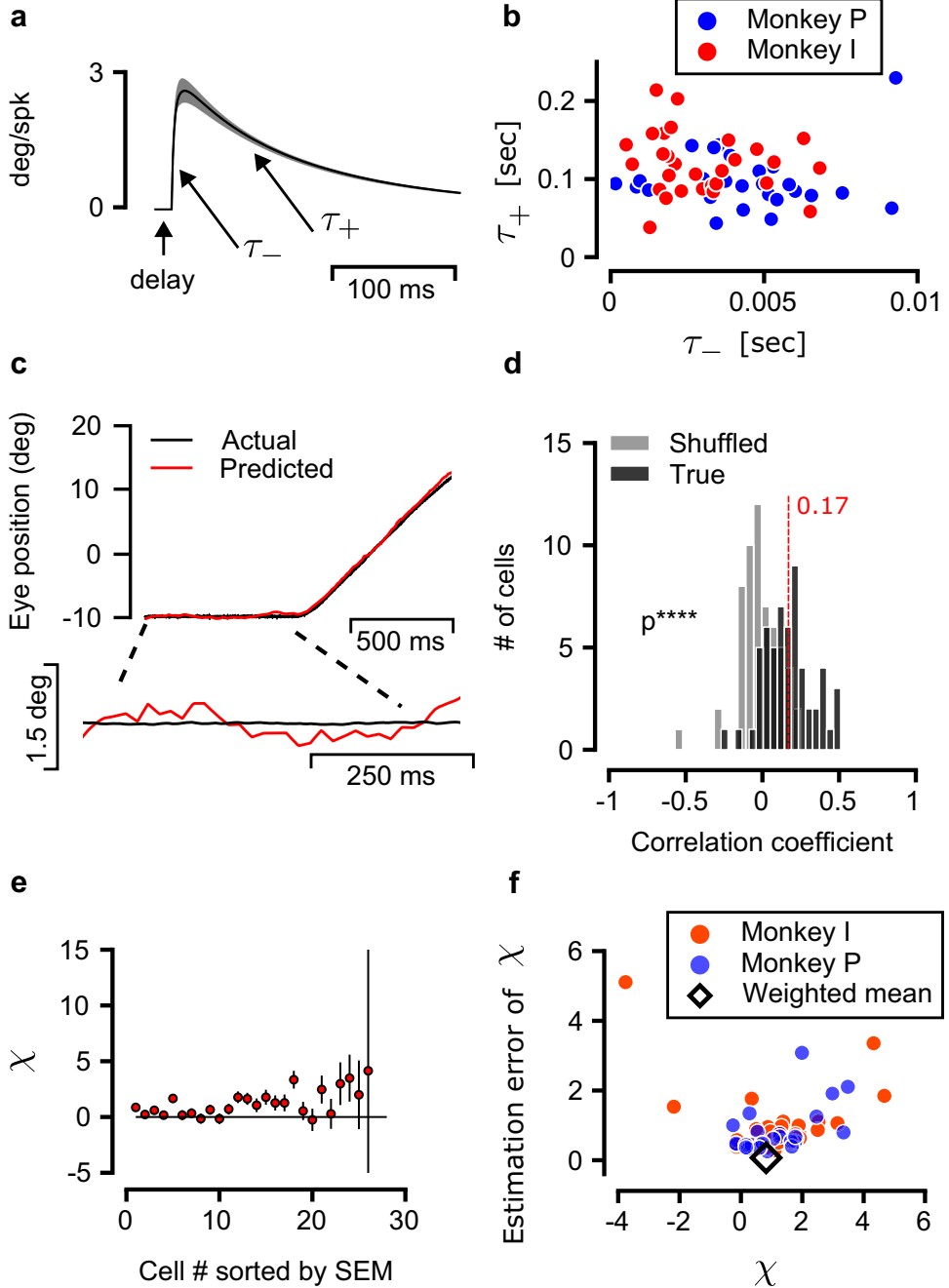

**Fig. 2 Predicting the eye trajectory from OMN spikes. a** Double exponent kernel as fitted from the data, black and gray traces represent the mean and SEM over all kernels optimized. **b** Fast ($\tau_- \approx 5$ ms) and slow ($\tau_+ \approx 120$ ms) time constants of the fitted kernels, shown for all analyzed OMNs (excluding one cell with $\tau_- = 0.01$ s and $\tau_+ = 0.79$ s). **c** Actual and predicted eye position during a single trial. Lower panel, zoom in on fixational segment. The predicted eye position, based on the spikes, was obtained using eqs. (4)–(7) (Methods). Note that the predicted eye position is much more variable than the actual eye position. **d** Distribution across OMNs of correlation coefficients between actual and predicted eye position differences across 350 ms fixational intervals (black). One sided $t$ test, $p = 3.8 \times 10^{-12}$ ($n = 57$ OMN cells. Number of fixational segments varied from cells to cell and was 302 on average. Histogram shown in Extended Data Fig. 7c). Gray, distribution of correlation coefficients obtained from each OMN after shuffling (see Methods). **e** For each OMN of monkey I we estimated $\chi$ from the data, which is the fraction of measured eye position variance that can be attributed to a central source upstream of the OMNs. Cells ($N = 26$ shown) are sorted along the horizontal axis by the estimation error of $\chi$. Circles represent the estimates of $\chi$ (eq. (15)). Error bars extend above and below the circles by $\Delta\chi$, where $\Delta\chi$ is the standard error of the estimate, as described in Methods. The horizontal black line is $\chi = 0$ for reference. Note that while $\chi$ is expected to lie within the range [0, 1], our estimates for this quantity based on eq. (15) are noisy, due to the OMN spiking noise and the limited number of trials for each cell. Therefore, these estimates can deviate from the expected range. For visual clarity, two cells with $\chi = 73 \pm 37$, $44 \pm 41$ are not shown. **f** Distribution of both $\chi$ (horizontal-axis) and its estimated error (vertical-axis), from OMNs recorded in the two monkeys. Black diamond: weighted average of the results from both monkeys (see also Methods) indicates that a large fraction in the variance is attributed to a central source, $\langle\chi\rangle = 0.82 \pm 0.07$ (weighted mean ± weighted error).

arise in the muscle or plant dynamics and includes also the noise in the measurement of the eye position. In addition, we assumed that the same linear relationship between OMN activity and eye position holds during large eye motions and during fixational drift. We thus used the estimator which was fit to predict large eye motions, and evaluated the covariance between the motion predicted by this estimator over 350 ms intervals of fixational drift, and the actual eye motion. The second assumption above implies that the central contribution to the output of the estimator is scaled correctly in units of eye position (yet may be swamped by the contribution of the intrinsic spiking noise). Thus, the covariance between the predicted motion and the measured eye motion extracts, from the overall variability in the measured eye motion, the variance that originates in the central source (see also Methods).

Using this approach, we estimated the fraction of the variability in eye motion which is driven by the common input in 350 ms intervals (denoted below by $\chi$), separately for each OMN. If we were to evaluate the covariance between the predicted and actual eye motion precisely, we would expect to obtain values of $\chi$ in the range [0, 1]. However, our estimates of the covariance were noisy since both the measured and predicted eye trajectories are influenced by sources of noise other than the common input (Methods) and the number of trials was limited. Hence, empirical values of $\chi$ obtained from individual OMNs could lie outside of the range [0, 1] (as corroborated also in simulated data, discussed below).

The estimates, from one monkey, are presented in Fig. 2e with their standard errors, and results from all OMNs (both monkeys) are shown in Fig. 2f. In Fig. 2f each symbol corresponds to one OMN, and the horizontal and vertical axes represent the estimate of $\chi$ and its standard error, correspondingly. An average of all these estimates for $\chi$, which takes into account the standard errors, produced a much tighter estimate than obtained from single OMNs: a fraction $\chi = 0.82 \pm 0.07$ (weighted mean ± weighted error, Fig. 2f and Extended Data Fig. 1) of the variance in eye motion during 350 ms intervals is driven by the common input to the OMNs. This central result of our work establishes that most of the eye motion during fixational drift originates in central neural circuits.

**Mean square displacement curves**. Even though our analysis above points to a central source for the motion upstream of the OMNs, we sought additional evidence that the correlations between eye motion and OMN activity are not simply a result of the spiking noise in the activity of single OMNs. Such evidence could be obtained by examining how the mean squared displacement (MSD) of the horizontal eye motion varies as a function of the time lag. In Fig. 3a we plot the MSD curve from the horizontal eye motion measurements of monkey P, calculated over all measured eye trajectories during fixation (black trace, see Methods). The MSD approaches a constant at zero time lag (Fig. 3a and Extended Data Fig. 2a) which is indicative of measurement noise with a variance of order $10^{-3}$ deg$^2$, in agreement with independent estimates of the measurement noise variance (Extended Data Fig. 2a, inset). The variance arising from the measurement noise was subtracted from the MSD measurements (see Methods and Extended Data Fig. 2) to obtain the MSD of the actual eye motion (blue symbols in Fig. 3a, b). The MSD curves measured in both monkeys (Fig. 3b) demonstrated super-diffusive statistics over the entire range of time lags that we examined: on logarithmic axes, the MSD increased steadily with a slope $1 < \alpha < 2$ (a slope $\alpha = 1$ characterizes Brownian motion, whereas a slope $\alpha = 2$ characterizes motion at constant velocity). In addition, the logarithmic slope decreased as a function of the

time lag. Both observations are in agreement with measurements in human subjects[9].

We next estimated the MSD curve of the eye motion that would arise from independent spiking variability of OMNs (see Discussion), driving the occular muscles (Methods). The estimated MSD was too small to account for the experimentally observed variability across all time lags (Fig. 3b), and at time lags exceeding ~0.1 s the predicted MSD was negligible compared to the experimentally observed MSD, in agreement with our previous conclusion that motion is driven upstream of the OMNs.

Furthermore, the predicted logarithmic slope of the MSD curve, generated by the OMN spiking noise, was small at all time lags compared to the measurements, and beyond a few hundred ms the predicted curve saturated, whereas the experimentally measured MSD curve continued to increase steadily (Fig. 3b). Similar saturation is expected to arise from any form of temporally uncorrelated noise which is fed into the muscle dynamics, at time lags exceeding the characteristic time scale of the muscle response (~180 ms, see Methods) . Thus, even if our estimates of the single OMN variability are incorrect, or if there is noise correlation between OMNs (which is unlikely, see Discussion), the outcome would be a vertical upward shift of the logarithmic MSD curve relative to the black trace (broken gray line, Fig. 3b), which cannot match the shape of the empirical curve. In conclusion, the steep logarithmic slope of the experimental MSD and its non-saturating behavior indicate that the input to the OMNs must itself be characterized by diffusive statistics, with a MSD curve that increases steadily as a function of the time lag, at least up to time lags of order 1 s. This conclusion provides an important insight into the possible underlying mechanism.

**Stochastic diffusion in a memory circuit**. OMNs receive their input from the oculomotor integrator, a memory network which is capable of holding the eyes still between saccades by providing steady input to the ocular muscles[36]. Since the horizontal eye position is a continuous variable, the oculomotor integrator maintaining horizontal gaze is commonly modeled as a continuous attractor neural network[30–32,43], whose dynamics are characterized by a one-dimensional manifold of semi-stable steady states. The different activity patterns of the network along the attractor are mapped, through the synaptic outputs to the OMNs, to different horizontal eye positions.

It has long been argued on theoretical grounds that in continuous attractor networks, neural noise can drive diffusive motion along the attractor, which gradually degrades the stored memory[33–35,44]. Fig. 4a schematically illustrates how the diffusive motion is generated: neural noise continuously perturbs the state of the network. While perturbations that shift the neural population activity away from the attractor decay rapidly, perturbations along the attractor do not decay since all positions along the attractor correspond to semi-stable states of the dynamics. These non-decaying perturbations cause random displacement in the position of the network along the attractor that accumulate over time with a variance that increases linearly with the time lag, giving rise to diffusive motion with simple random walk statistics along the attractor manifold[33,35]. Successes in directly observing this diffusive motion, and especially in relating it to neural mechanisms in specific brain circuits have been scarce and incomplete[45,46]. Diffusive dynamics in the output of the oculomotor integrator has the potential to drive ocular motion with a nonsaturating MSD curve. For this reason, we hypothesized that fixational drift is driven by diffusion within the oculomotor integrator. To put this hypothesis to quantitative test

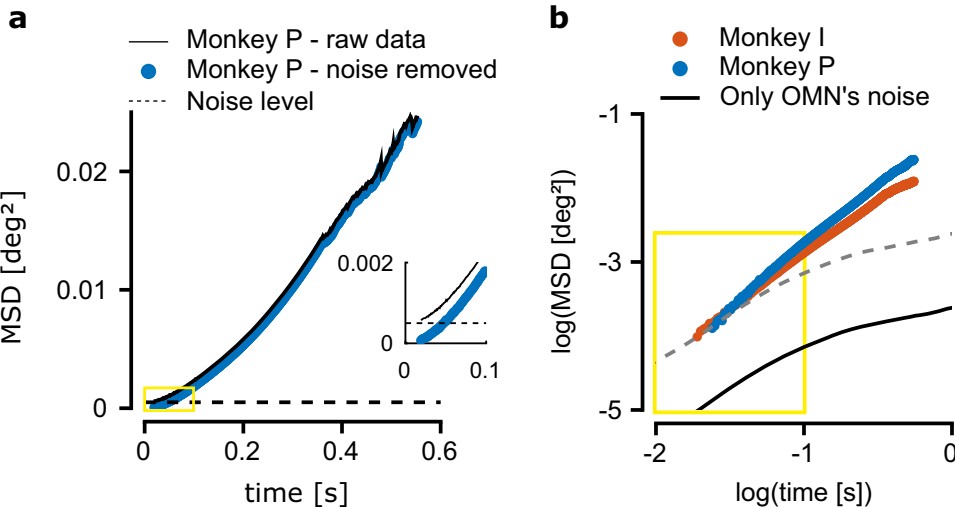

**Fig. 3 Motoneuron variability can't explain the mean squared displacement at large time lags. a** Mean squared displacement (MSD) curve, measured for monkey P with measurement noise (black trace). Dashed horizontal line: estimate of the contribution arising from measurement noise (Methods). Blue circles: MSD after subtraction of the measurement noise contribution. Inset: zoom in over the area marked by the yellow rectangle. Note that the black trace doesn't approach zero at zero time lag, which is indicative of contamination by temporally uncorrelated measurement errors. **b** Mean squared displacement (MSD) curves from both monkeys (blue and red traces), using logarithmic scales in both axes, after subtraction of the measurement noise contribution. Black trace: MSD obtained by simulation of the contribution arising from the spiking variability of OMNs, filtered by the response dynamics of the muscles and mechanics of the eye (assuming 2000 heterogeneous OMNs, see Methods). The black trace flattens at time lags exceeding ~1 s and can only account for ~1% of the total variance at a time lag of ~900 ms. Dashed gray trace: translated copy of the black trace along the logarithmic vertical axis demonstrates that the slope of the MSD curve generated by the OMN variability cannot match the slope of the measured MSD curve. The yellow marked rectangle represents the same range of time lags and MSDs as the yellow rectangle in (**a**), on the logarithmic axes.

we adapted a model of the goldfish oculomotor integrator[31] in two ways: first, we replaced the rate model by a spiking model in which single neuron activity is variable. Second, we adapted the model to the parameters of the primate visual system (see Methods). Specifically, key parameters that affect the diffusivity[33] such as the number of neurons, their tuning curves, and their spiking variability, were set based on experimental estimates in Macaque (Methods). With these parameters, the output of the network exhibited random diffusion (example trajectories shown in Fig. 4c), with a slope of the MSD curve (using logarithmic axes) close to unity (Fig. 4d). The mean square displacement over a time lag of 350 ms was in the order of 0.1 deg², comparable to the typical range of fixational drift.

To relate this result more quantitatively to the statistics of fixational drift, we constructed a mathematical model of central and peripheral contributions to the eye dynamics: the stochastic dynamics of the oculomotor integrator, OMNs, muscles and the ocular plant. We also incorporated in the model a visual feedback mechanism, which can partly correct for the motion of the target due to drift but involves a relatively long delay due to the likely involvement of the cortical or sub-cortical areas[9,21,47] (Fig. 5a, schematic of the model). The output of the integrator was used as an input to a diverse population of OMNs innervating different extra-ocular muscle fibers, with intrinsic spiking noise[48] and with dynamic impact on the muscles which was modeled based on[37] to determine the horizontal eye position.

Numerical simulations of the model using biologically plausible parameters (Methods) generated eye motion that resembles measured eye trajectories (Fig. 5b). The relationship between the output signal of the oculomotor integrator and the actual eye position is demonstrated in Fig. 5c: the oculomotor integrator output is noisy due to the spiking variability of oculomotor integrator neurons, whereas the actual eye trajectory is smoother, due to the mechanics of the muscles and the eye ball. When examined at a coarse scale of hundreds of milliseconds, the eye position follows the output of the oculomotor integrator,

indicating that the diffusive dynamics within the integrator dominates the statistics of the final eye position at this scale. Importantly, the simulated MSD curve of the eye position (black trace, Fig. 5d) increases steadily as a function of the time lag at time scales exceeding ~100 ms due to the diffusive dynamics of the oculomotor integrator output, and explains well the MSD statistics which were measured from the data, both in terms of its slope and amplitude (magnitude of the variance).

Since the motion is primarily driven by the output of the oculomotor integrator, its amplitude is determined mostly by the parameters of the oculomotor integrator network. As shown above, the oculomotor integrator output follows approximately the statistics of a simple random walk, which can be characterized by a a single parameter, the diffusion coefficient $\mathcal{D}$. It can be shown[33] that $\mathcal{D}$ scales with the parameters of network as

$$\mathcal{D} \propto (\mathrm{CV})^2 N^{-1} \lambda_0^{-1} k_0^{-2} \tau_s^{-2} \tag{1}$$

where $N$ is the number of neurons in the oculomotor integrator network, CV is their spiking coefficient of variation, $\tau_s$ is the synaptic time constant, and $k_0, \lambda_0$ are the typical values of the position sensitivity and discharge rate within the population of oculomotor integrator neurons. Since several of the parameters in eq. (1) are not known precisely, there is some freedom in adjusting the diffusion coefficient to match the experimental data by tuning the model parameters, and there is no unique way to achieve a good match. Nevertheless, it is noteworthy that the model accounts well for the amplitude of fixational drift, with parameters whose order of magnitude is correct.

While there is some freedom in shifting the predicted MSD curve along the vertical axis (in logarithmic scales) by adjusting $\mathcal{D}$, there is much less freedom in adjusting the MSD curve in other ways. The logarithmic slope of the MSD curve is primarily determined by the muscle response dynamics, whose modeling is firmly based on experimental data. The muscle response dynamics, acting on the diffusive input from the oculomotor integrator, results in superdiffusive motion with a logarithmic

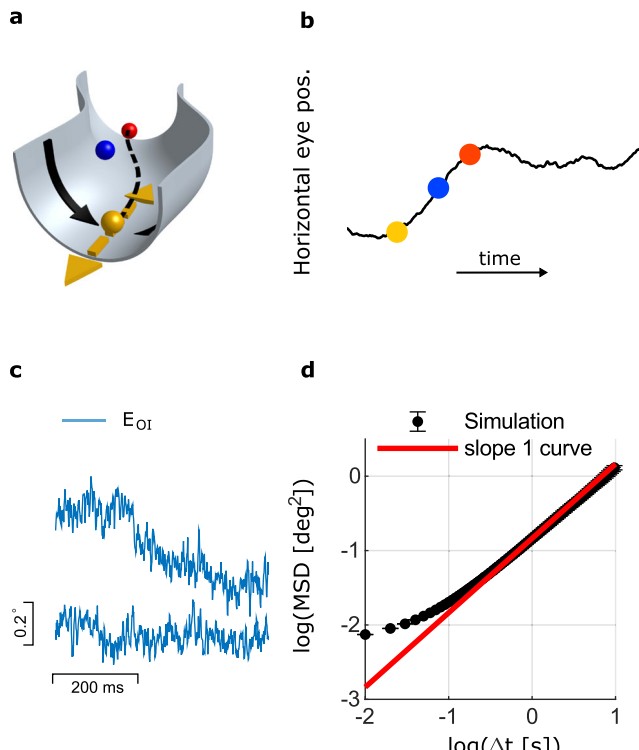

**Fig. 4 Effect of intrinsic noise in a continuous attractor neural network. a** Schematic illustration of a continuous attractor manifold. Steady states of the dynamics can be conceptualized as minima of an energy function defined in the neural state space[30]. For visualization purposes, two dimensions of the neural state space are depicted in the plane, and the vertical dimension represents the energy. Each point along the curved valley (dashed line) maps to a specific horizontal eye position. During fixation the state of the network is set initially to represent a particular eye position (yellow circle). Noise in the activity of neurons within the network dynamically perturbs the state of the network. Such perturbations generally decay (black arrows), but perturbations along the flat direction (dashed yellow arrows) do not decay, causing motion along the valley that builds up over time in the form of diffusive motion[33]. The state of the network at two time points after initialization is depicted by the blue and red balls. The projection on the dashed line determines the horizontal eye position (**b**). **b** The horizontal eye trajectory resulting from the noise-driven diffusive motion. **c** Examples of two fixational eye trajectories that emerge from the simulation of the oculomotor integrator model network, with neural noise. **d** The MSD statistics of 100 eye trajectories as in (**c**). At sufficiently large time lags the motion follows the statistics of simple diffusion, with a logarithmic slope equal to 1 (red line).

slope that decreases with increase of the time lag, and the predicted dependence of the logarithmic slope on the time lag matches the data very well. Thus, key features of the statistics are explained well by the model, independently of fitting parameters.

The main consequence of the visual feedback mechanism in the model is to decrease the logarithmic slope of the MSD curve at large time scales, which is otherwise a bit too steep compared to the experimental measurements (Extended Data Fig. 3a, b. See also SI Notes). On the other hand, the visual feedback mechanism has little influence on the slope of the logarithmic MSD curve at shorter time scales (Extended Data Fig. 3c), and is unlikely to affect this feature due to the synaptic delays involved. Therefore, we propose that the reason for the super-diffusive statistics of fixational eye drifts lies in the diffusive dynamics within the oculomotor integrator (combined with the muscle dynamics), and is largely independent of visual feedback mechanisms. On the

other hand, visual feedback can modulate the statistics of the motion, especially over long time lags, in accordance with the observations that these statistics are influenced by the visual task[4,14,21–25].

In summary, with a few parameters – most of which were chosen based on known features of the primate oculomotor system (Methods), the model produces simulated MSD curves that match experimental observations very well, both in magnitude and in shape (Fig. 5d and Extended Data Fig. 5).

Finally, the availability of a generative model of fixational drift and the accompanying neural activity, offered an opportunity to test the methodology that was used in Fig. 2e, f to identify the central contribution to the motion. We generated a data set of simulated eye trajectories and OMN spikes, and analyzed this data using the methodology that we previously applied to the experimental data. The numbers of analyzed OMNs and trials were similar to the experiment (Methods). Estimates for $\chi$ (Fig. 5e) were highly variable across the simulated OMN population, in similarity to the results obtained in Fig. 2f, yet their weighted average pointed to the existence of a central source to the motion. In an alternative model, in which the motion was fully driven by independent noise in the OMNs, the same methodology pointed correctly to the lack of a central contribution to the motion (Extended Data Fig. 4). The simulated data produced also a similar distribution of correlation coefficients between eye motion and its prediction based on single OMN spikes, as in the analysis of the experimental data (Fig. 5f).

## Discussion
We showed for the first time that fixational drift is correlated with neural activity in the oculomotor control circuitry of the brainstem, and identified the main source of the motion in central neural circuitry upstream of the OMNs. The statistics of the motion provided an important clue on the identity of the upstream drive, pointing to noise-driven diffusion in the oculomotor integrator as a likely source of the motion. Theoretical modeling, constrained by the physiology of the primate oculomotor system, provided further support to this hypothesis since it accounted for the magnitude and detailed statistics of the motion. Taking the view that the oculomotor integrator is a short-term memory network[30–32], we thus propose that fixational drift offers direct observation of diffusive dynamics in continuous attractor networks, and an arena for probing mechanistically how noise affects the storage of continuous parameter memory in neural circuits.

Both in our analysis of the OMN recordings and in our theoretical modeling of contributions to the MSD curves, we evoked the assumption that the intrinsic spiking is independent in different OMNs. This assumption is based on the anatomy of the oculomotor nuclei that control the horizontal position of the eye, and on stimulation studies: anatomical tracing studies did not find axon collaterals originating from OMNs in the abducens nucleus (from which we recorded)[49,50], indicating that these neurons only innervate the lateral rectus muscle. In the medial rectus axon collaterals do not terminate within the nucleus[51–53], indicating that there are no local recurrent connections, and in both areas stimulation studies did not find evidence for recurrent excitation of inhibition between OMNs[54]. Therefore, it is unlikely that intrinsic spiking noise is spatially correlated among different OMNs. Furthermore, the temporal time course of eye motion in response to stimulation of OMNs matches the time course of the muscle response[55,56], which excludes the existence of additional reverberation arising from recurrent connectivity within the oculomotor nucleus.

Both intrinsic variability and noisy inputs can drive stochastic diffusion in continuous attractor networks. We assumed that the

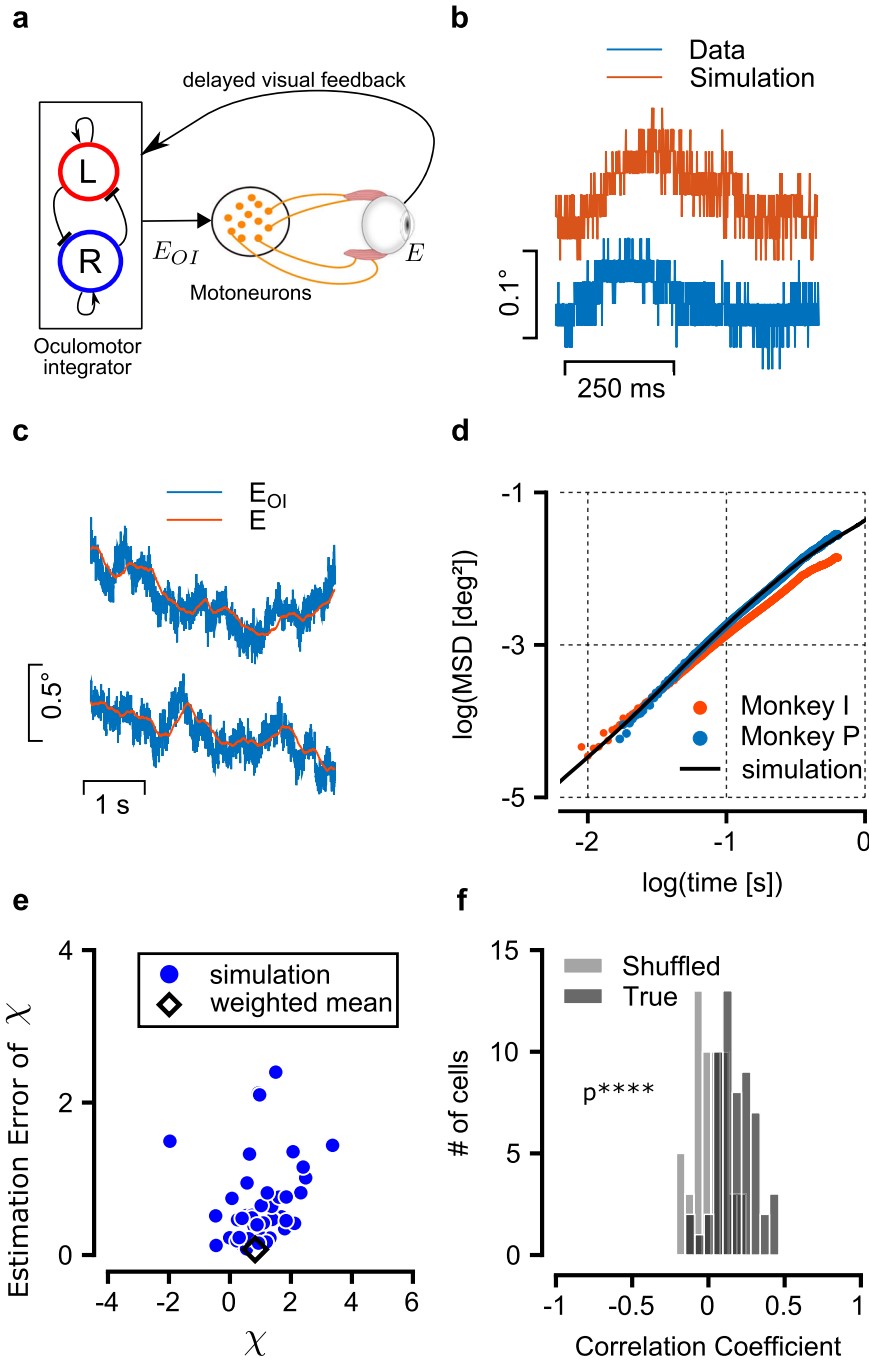

**Fig. 5 The model. a** Schematic illustration of the model: an internal representation of the eye position, $E_{OI}$ is maintained by the oculomotor integrator and transmitted to the OMNs. OMNs activate the extraocular muscles, resulting in the actual eye position, $E$. The actual eye position serves as a delayed visual feedback to the oculomotor integrator. **b** Example of simulated and measured fixational eye trajectories. The model generates eye trajectories which are qualitatively similar to the recordings (digitization applied to the simulated trajectory to facilitate the comparison). **c** Simulated trajectories of $E_{OI}$, the internal eye representation (blue), and of $E$, the corresponding actual eye position (red). **d** Mean squared displacement (MSD) curves, shown using logarithmic axes, as measured from the eye trajectory recordings (blue and red traces) and from simulation of the computational model in (**a**) (black). Panels (**e**–**f**), Application of similar methodology as in Fig. 2 to simulated data generated by the model. The number of OMNs (57) and trial numbers per OMN (100) were set in similarity to the experimental data. **e** Distribution of estimates for $\chi$ and their standard errors (similar to Fig. 2f). Estimates for $\chi$ are highly variable, in similarity to those obtained from the experimental data, due to the single OMN noise. Consequently, individual estimates for $\chi$ can be negative or exceed unity. The weighted estimate of $\chi$ from all OMNs, $0.82 \pm 0.07$, (black diamonds) indicates that the variance of the motion is dominated by the central drive. In an alternative model, where the dominant contribution to the motion is peripheral, the estimate of $\chi$ obtained using the same method is close to zero (Extended Data Fig. 4). **f** Distribution of correlation coefficients between eye displacement over 350 ms intervals, and their estimates based on the spikes of individual OMNs, with and without shuffling (similar to Fig. 2d). The mean value of the distribution, $\langle R \rangle = 0.17 \pm 0.02$, significantly deviates from zero (one sided $t$ test, $p = 10^{-14}$, shuffled distribution shown in gray).

noise is intrinsic to neurons within the oculomotor integrator network, and this assumption accounted well for the amplitude of the motion. However, we only have approximate estimates for some parameters that influence the diffusion coefficient $\mathcal{D}$ of the oculomotor integrator output, such as the number of neurons in the network. Hence, we cannot rule out that the diffusive dynamics in the state of the oculomotor integrator is driven in part by noisy premotor inputs to this network, such as those arising from vestibular, optokinetic, and vergence signals.

Our focus on noise in the occulomotor integrator as a drive of diffusive input into the OMNs does not imply that fixational drift is unaffected by additional mechanisms, acting within the visual-motor pathway, such as fatigue[57,58] and feedback mechanisms. The incorporation of a visual feedback mechanism in our model demonstrates, indeed, that a visual feedback loop can modulate the statistics of the motion at time lags exceeding ~100 ms – but is unlikely to influence the slope of the MSD curve at shorter time scales, due to the synaptic processing delays involved, of order 60–80 ms[9,21,59]. We note also that the strength and dynamics of modulation by visual feedback are likely affected by the salience and structure of the visual stimulus, in accordance with the observations that the detailed statistics, but not the superdiffusive nature, of fixational drift is influenced by the visual task[4,10,14,21–25]. It has been recently argued[27] that the superior colliculus (SC) may play a key role in the modulation of fixational drift in response to visual inputs, for several reasons: first, neural responses to fixational eye drifts, which likely arise from sensory inputs, were observed in superficial layers of the SC[60]. Second, the SC is involved in control of eye motion during large eye motions[61] and microsaccades[29]. Finally, neural activity in the SC has been shown to represent a desired eye position during smooth eye motions[62].

Finally, fixational drift is highly consequential for visual perception in the fovea, even though its amplitude is tiny compared to saccades[4,10,12–15,17,18]. Detailed understanding of the mechanisms underlying fixational drift is likely to advance the research on its functional consequences, by opening up the possibility to examine whether specific parameters of the oculomotor pathway, such as the variability of neural activity within the oculomotor integrator and the dynamics of the muscle response, are tuned to optimize visual function.

## Methods

**Behavioral task and recordings**. We have reanalyzed data reported in published studies[38,63,64]. Data were collected from two male rhesus macaque monkeys (Macaca mulatta). All procedures had been approved in advance by the Institutional Animal Care and Use Committee at UCSF, where the experiments were performed. Procedures were in strict compliance with the National Institutes of Health Guide for the Care and Use of Laboratory Animals. To instrument monkeys for experiments we implanted a coil of wire on one eye[65] and hardware to allow us to restrain the monkey's head. After monkeys had recovered from surgery, we trained them to sit in a primate chair with the head restrained, and to fixate and track spots of light that moved across a video monitor placed in front of them. In a later surgery, we used a trephine to make a hole in the skull, and secured a recording cylinder aimed at the brainstem[66]. During experiments, we lowered glass-coated platinum-iridium electrodes into the brainstem to record from neurons in the abducens nucleus. The abducens nucleus was distinguished by the characteristic singing activity associated with ipsiversive eye movements. We identified OMNs, as opposed to internuclear neurons (INNs) based on the criteria used in[67], Extended Data Fig. 6. Voltage waveforms from the electrodes were amplified and bandpass filtered, usually between 500 Hz and 5 kHz. We sampled the voltage waveforms from the electrode continuously at 25 kHz to allow off-line spike sorting (Plexon and custom software). Our analysis requires exquisite sorting as we are looking for correlation between small modulations in rate and low amplitude eye movements. We therefore corrected for errors in sorting by looking for discontinuities in the firing rate of the neurons. The high regularity of the abducens neuron makes is possible to detect potential missed or erroneously added spikes. We then inspected the corresponding recorded waveform and corrected if necessary. Visual stimuli appeared on a monitor at a distance of 30 cm from the monkey's eye. Targets were bright 0.6° circles on a dark background. We recorded neural activity during pursuit of step-ramp target motions[68]. At the start of each

trial, the monkeys had a second to acquire fixation on a stationary target. They were then required to fixate for an additional 500–700 ms within a 2°–3° square window. The target then displaced to a location eccentric to the position of gaze (step), and immediately began moving toward the fixation point (ramp). The size of the displacement was chosen to minimize the presence of initial saccades and hence, varied slightly between monkeys, recording days and target speeds.

**OMN firing rate estimators**. We obtained estimators $\hat{f}(t)$ of the OMN firing rates based on the eye trajectory as follows. The estimators were expressed as a linear combination of the eye position, velocity, and acceleration[36,69–71]:

$$\hat{f}(t) = k\left[E(t + \Delta t) - E_T\right] + r\dot{E}(t + \Delta t) + m\ddot{E}(t + \Delta t) \quad (2)$$

where the parameters $k$, $r$, $m$, $\Delta t$, and $E_T$ were chosen to fit simultaneous measurements of the OMN firing rate during the full extent of each trial, including the fixational and smooth pursuit components. The firing rate $f(t)$ was extracted from the spike train as the inverse of the inter spike interval[36], and was thus taken to be constant between successive spikes. We discarded trials in which the measured firing rate went below 20 Hz[37], since eq. (2) only holds above the OMN threshold. The parameters were chosen to minimize the loss function

$$L = \sum_{i=1}^{N} \sum_{t_j=0}^{T_i} \left[f_i(t_j) - \hat{f}_i(t_j; k, r, m, \Delta t, E_T)\right]^2 \quad (3)$$

where $T_i$ is the duration of the $i$-th trial, $t_j \in [0, T_i]$ are discrete time samples within a trial, and $N$ is the total number of trials. To generate an estimate to the local eye acceleration we smoothed the velocity signal using a Savitzky-Golay filter of order 3 and 21 ms length[72] and performed a two-sided numerical differentiation. To generate a firing rate prediction for each of the hundreds of trials per cell we followed a "leave one out" scheme: we fitted the cell parameters to all but one trial, and used these parameters to generate the firing rate prediction for the left out trial.

**Extracting the fixational segments**. Microsaccade onsets and endings were detected using the eye motion in 2d (horizontal and vertical components), by applying a combination of thresholds for the magnitude of the velocity (10 deg/sec) and the magnitude of the acceleration (1000 deg/sec²). The horizontal position fixation segments used for further analysis started 30 ms after termination and ended 30 ms before initiation of the microsaccades. Finally we verified manually that the fixational segments didn't contain any microsaccades. A histogram of the fixational segment durations is shown in Extended Data Fig. 7.

**Firing rate correlation coefficients**. Correlation coefficients in the firing rate representation (Fig. 1) were calculated by generating two vectors of the actual and predicted firing rates from all trials available per cell. Each vector was generated as follows: for each fixational segment we calculated the firing rate, either actual or predicted, and substracted from it the mean firing rate during that fixation. Finally we concatenated contributions from all fixational segments and calculated the Pearson correlation between the predicted and actual firing rate vectors.

**Model inversion**. Eye position was predicted from the spiking activity by causal filtering:

$$\hat{E}(t) = E_T + (h * \xi)(t) \quad (4)$$

Here, $\xi(t)$ is the spike train emitted by the cell:

$$\xi(t) = \sum_i \delta(t - t_i) \quad (5)$$

where $t_i$ is the timing of the $i$-th spike and $h(t)$ is the kernel of the linear filter that represents the inverse of the relationship in eq. (2):

$$h(t) = \frac{\Theta(t - \Delta t)}{m} \frac{\tau_+ \tau_-}{\tau_+ - \tau_-} \left[e^{-(t-\Delta t)/\tau_+} - e^{-(t-\Delta t)/\tau_-}\right] \quad (6)$$

where

$$\tau_{\mp} = 2\left(\frac{r}{m} \pm \sqrt{\left(\frac{r}{m}\right)^2 - 4\frac{k}{m}}\right)^{-1} \quad (7)$$

and $\Theta$ is the Heaviside step function.

**Eye position correlation coefficients**. Correlation coefficients between the measured eye position and its prediction based on single OMN spike trains (Fig. 2d) were evaluated as follows. First, eq. (4) was used to generate a prediction $\hat{E}(t)$ of eye position. We discarded fixational segments shorter than 350 ms and broke longer segments into non overlapping 350 ms fixational segments. For each fixational segment we calculated the difference in the measured eye position, $\delta E = E(t = 350 \text{ ms}) - E(t = 0 \text{ ms})$, across the fixational segment, and the corresponding prediction $\delta \hat{E} = \hat{E}(t = 350 \text{ ms}) - \hat{E}(t = 0 \text{ ms})$. Finally, we calculated the Pearson

correlation coefficient between all predicted and measured eye position differences

$$R = \frac{\text{Cov}(\delta E, \delta \hat{E})}{\sqrt{\text{Var}(\delta E)\text{Var}(\delta \hat{E})}} \qquad (8)$$

In addition we calculated the Spearman rank correlation between these two sets of eye differences which also demonstrated significant correlation ($p = 2.5 \times 10^{-12}$, Extended Data Fig. 8).

In Fig. 2d the shuffled distribution (gray) was obtained by randomly associating, for each cell, the measurements of $\delta E$ and of $\delta \hat{E}$ from different fixational segments. Note that in the shuffled data set there is no underlying correlation between $\delta E$ and $\delta \hat{E}$. Accordingly, the distribution of correlation coefficients is centered around zero, yet its width is similar to the histogram that was obtained without shuffling, in agreement with the interpretation that the variability in measurements of $R$ (reflected in the width of the distribution) is due to the estimation error of $R$.

**Standard error of the estimates for the covariance and R.** The standard error of the covariance estimate (used to calculate the errors in Fig. 2e, Extended Data Fig. 1) is given[73] by

$$\left(\Delta \text{Cov}(\hat{E}, E)\right)^2 = \frac{1}{n}\left[d(\hat{E}, E) + \frac{1}{n-1}\text{Var}(E)\text{Var}(\hat{E}) - \frac{n-2}{n-1}\text{Cov}^2(\hat{E}, E)\right]$$
$$d(\hat{E}, E) = \left\langle \left[(E - \langle E \rangle)(\hat{E} - \langle \hat{E} \rangle)\right]^2 \right\rangle \qquad (9)$$

where $n$ is the number of fixational segments used to evaluate the covariance.

The standard error of the estimate for $R$, eq. (8) was evaluated as follows:

$$(\Delta R)^2 = R^2\left[\left(\frac{\Delta \text{Cov}(\hat{E}, E)}{\text{Cov}(\hat{E}, E)}\right)^2 + \left(\frac{\Delta \text{Var}(E)}{\text{Var}(E)}\right)^2 + \left(\frac{\Delta \text{Var}(\hat{E})}{\text{Var}(\hat{E})}\right)^2\right] \qquad (10)$$

where $\Delta \text{Var}(E)$ and $\Delta \text{Var}(\hat{E})$ are the standard errors of the estimates for $\text{Var}(E)$ and $\text{Var}(\hat{E})$, obtained using expressions similar to eq. (9).

**Estimated contribution of central source to measured eye position.** To estimate the contribution of a central source to the measured eye position, we evoke the following assumptions. We assume that the measured eye position $E$ can be written as:

$$E = E_C + \eta \qquad (11)$$

where $E_C$ is a contribution arising from sources upstream of the OMNs, and $\eta$ represents contributions arising from the OMNs as well as other sources of noise downstream of the OMNs. We assume that $\eta$ and $E_C$ are uncorrelated.

Similarly, we assume that the estimate of eye position $\hat{E}$, obtained from the spike train of a single OMN can be written as

$$\hat{E} = E_C + \epsilon \qquad (12)$$

where $\epsilon$ represents the error of the estimator and is assumed to be uncorrelated with $E_C$. Thus, we assume that the estimator, which was fitted to measurements obtained during large eye motions, remains unbiased during fixational intervals. Note that during fixation the variance of $\epsilon$ is large compared to that of $E_C$, which is the main reason why the correlation coefficient between $\hat{E}$ and $E_C$ is small compared to unity.

Our goal is to estimate what fraction of the variance in the measured eye position, $E$, is due to the central source, $E_C$, over a fixational segment (in our analysis we took segments of duration 350 ms):

$$\chi = \frac{\text{Var}(\delta E_C)}{\text{Var}(\delta E)} \qquad (13)$$

where $\delta x$ represent the difference between the value of a variable $x$ at the end of the interval and its value at the start of the interval. Under the assumption that the noise terms $\eta$ and $\epsilon$ are uncorrelated,

$$\text{Cov}(\delta \hat{E}, \delta E) \approx \text{Var}(\delta E_C) \qquad (14)$$

In fact, there is a very weak correlation between $\eta$ and $\epsilon$ which arises from the fact that $\eta$ includes the noise contributions from all the OMNs, and $\epsilon$ represents the contribution of noise to the estimate error from a single OMN. This correlation however, is of order $1/N_m$ where $N_m$ is the number of OMNs, and we can safely neglect it, as corroborated also from our separate estimate of the OMN contribution to fixational drift (Fig. 3). Therefore, our estimate of $\chi$ is given by

$$\chi \approx \frac{\text{Cov}(\delta \hat{E}, \delta E)}{\text{Var}(\delta E)} \qquad (15)$$

The standard error in our estimate of this quantity is dominated by our ability to measure the covariance from a limited number of trials. The estimated errors in Fig. 2d were thus calculated as

$$(\Delta \chi)^2 \approx \left(\frac{\Delta \text{Cov}(\delta \hat{E}, \delta E)}{\text{Var}(\delta E)}\right)^2 \qquad (16)$$

Note that ideally in eq. (16) we would like to include in the denominator the variance of the true eye position and not the measured eye position. However over intervals of duration 350 ms the measurement error of the change in the eye position is negligible compared to the actual motion, and is inconsequential for our estimate of the relative contribution of the central source to the motion.

Assuming that the errors in our estimates of $\chi$ are normally distributed, we calculated a weighted mean of this quantity across all cells from both monkeys as follows

$$\hat{\chi} = \sigma_\chi^2 \sum_i \frac{\chi_i}{\Delta \chi_i^2}$$
$$\sigma_\chi^2 = \left[\sum_i \frac{1}{\Delta \chi_i^2}\right]^{-1} \qquad (17)$$

where $\chi_i$ and $\Delta \chi_i$ are the estimated value and the estimated standard error of $\chi$ from the $i$'th cell. We used a weighted mean, and not a simple average over all cells, because of the large variation in $\Delta \chi$ across cells, which was due to differences in various factors such as the number of trials and the structure of the tuning curves.

**Empirical MSD curves**

*Calculation of empirical MSD.* The horizontal MSD, as presented in Figs. 3 and 5d, was calculated from the data in the following way: first we extracted all the inter-saccade eye trajectories as described above (Extended Data Fig. 4). Since each of the inter-saccade trajectories is of different length, we calculated the MSD separately in each fixational interval. In the second step we averaged, for each time lag, over all the available MSD measurements that correspond to this time lag, in all of the fixational intervals. Since the temporal resolution of the sampled eye trajectories is 1ms, the time lags at which the MSD is evaluated are integer multiples of 1 ms.

*Removal of measurement noise.* The measurements of the eye position from the search coil are corrupted by a measurement error of variance $\sim 10^{-3}$ deg$^2$. This is evident when plotting the MSD curve of the eye movements, where the measurement error gives rise to a flat MSD curve (when using logarithmic axes) up to time scales in which the diffusion becomes dominant. A similar estimate for the measurement noise was obtained by recording from a coil fixed in space (Extended Data Fig. 2, inset). Therefore, in Figs. 3, 5d and Extended Data Fig. 5 we subtracted the variance of the measurement noise to recover the MSD statistics of the eyes themselves. The MSD curve prior to noise subtraction is shown in Fig. 3a, and Extended Data Fig. 2. Mathematically if we denote by $E(t), \eta_m(t)$, the horizontal eye position and the measurement noise accordingly, then under the assumption that the measurement noise is white, the empirical MSD is given by

$$\left\langle \left[E(t + \Delta t) + \eta_m(t + \Delta) - E(t) - \eta_m(t)\right]^2 \right\rangle = \left\langle \left[E(t + \Delta t) - E(t)\right]^2 \right\rangle$$
$$+ \left\langle \left[\eta_m(t + \Delta) - \eta_m(t)\right]^2 \right\rangle \qquad (18)$$

$$= \langle [E(t + \Delta t) - E(t)]^2 \rangle + 2\text{Var}(\eta_m) \rangle \qquad (19)$$

therefore eq. (19) justifies that the estimated MSD curve, as shown in Figs. 3b, 5d and Extended Data Fig. 5, are a good approximation for the underlying MSD curve of the true eye position.

*MSD curve of low-pass filtered white noise.* Here we briefly discuss why any source of white noise injected to the OMNs will result in a saturating MSD curve, over time lags greater than the typical time scale of the filter. For simplicity assume that the eye position is given by white noise $\xi$, convolved with a low-pass filter $h(t)$. We assume that the white noise has zero mean and variance

$$\langle \xi(t)\xi(t') \rangle = A\delta(t - t') \qquad (20)$$

The MSD is then given by

$$\langle [E(t + \Delta t) - E(t)]^2 \rangle = 2\text{Var}(E(0)) - 2\langle E(t + \Delta t)E(t) \rangle \qquad (21)$$

where

$$\langle E(t + \Delta t)E(t) \rangle = A \int_0^\infty h(t)h(t + \Delta t)\,\mathrm{d}t \qquad (22)$$

Hence, the second term on the right hand side of eq. (21) decays to zero (and consequently the MSD saturates) when $\Delta t$ is large compared with the characteristic time scale over which the filter $h$ decays. For examples if $h(t) = \exp(-t/\tau)\Theta(t)$, we will get $\langle E(t + \Delta t)E(t) \rangle \sim \exp(-|\Delta t|/\tau)$.

**Model of fixational drift.** Our model of fixational drift (Fig. 5a) consists of three stages. First, an internal signal of the desired eye position is generated and held in the memory network of the oculomotor integrator. In the second stage this signal is conveyed to a population of spiking OMNs. The synaptic output of each OMN is translated into an actual eye position using eq. (2). Finally, The actual eye position is taken to be an average over all OMNs.

*Oculomotor integrator network.* The oculomotor integrator network model is based on the model proposed in[31] for the neural network that determines the horizontal

eye position in goldfish, adapted to the parameters of the primate oculomotor system. Briefly, the network consists of two populations, of which one is more active when the eye is directed to the left and the other is more active when the eye is directed to the right. Each population forms excitatory synaptic connections to itself and inhibitory synaptic connections to the other population. The connectivity is tuned such that there is a continuum of steady states, representing a continuum of stable eye locations, and such that the firing rates of the single neurons fit the experimentally observed tuning curves across the full range of eye positions.

We adapted the model of[31] in the following ways. First, we introduced noise by using spiking neurons, with a CV of the inter spike intervals of ~0.22[38,74] as described below. The oculomotor integrator network dynamics are thus described in our model by the following eqs.

$$\lambda_i^R(t) = \left[ \zeta_i^R E_{OI}(t) + \lambda_i^0 + \zeta_i^R F(t) \right]_+ \tag{23}$$

$$\tau_s \frac{d}{dt} S_i^R(t) = -S_i^R(t) + X_i^R \tag{24}$$

$$E_{OI}(t) = \sum_i \eta_i \left[ S_i^R(t) - S_i^L(t) \right] \tag{25}$$

$$X_i^R(t) = \frac{\xi_i^R(t)}{\lambda_t + \lambda_i^R(t)} \tag{26}$$

In eq. (23) $\lambda_i^R$ is the firing rate of neuron $i$ from the right population, and $[x]_+ = \max(x, 0)$. In eqs. (24), (25) $S_i^R$ is the synaptic output generated by neuron $i$ of the right population. We similarly denote by $\lambda_i^L$ and $S_i^L$ the firing rate and the synaptic output of neuron $i$ from the left population (see below). The spike train of neuron $i$ from the right population is denoted in eq. (26) by $\xi_i^R(t)$, and $\tau_s$ in eq. (24) represents the characteristic time scale of post-synaptic currents. As in[31], the influence of spikes on the synaptic current is nonlinear, through the relationship between $\xi_i^R$ and $X_i^R$, eq. (26), where $\lambda_t$ is a characteristic firing rate, in our case $\lambda_t = 60$ Hz. Note that if $\xi_i^R$ is replaced in eq. (26) by $\lambda_i^R$, the dynamical eqs. become identical to those of[31]. In eq. (23) $F(t)$ represents a visual feedback signal, which is determined as described below.

The variable $E_{OI}(t)$ appearing in eq. (23) represents an internal readout of the eye position from the neural activity, which is a linear function of the synaptic activities with weights $\eta_i$. This quantity also serves as the eye position readout from the network. The slope of the tuning curve of neuron $i$ from the right population is denoted by $\zeta_i^R$ and is positive, and $\lambda_i^0$ is an offset. In those neurons that are above threshold at central gaze (i.e. eye position zero), $\lambda_i^0$ is equal to the firing rate at central gaze.

For the left population, eqs. analogous to (23), (24), and (26), are obtained by replacing the superscripts $R$ and $L$. We assumed that for each neuron in the right population there is a matching neuron in the left population whose tuning curve is identical up to reversal of sign of the eye position. Hence, the slopes of the tuning curves in the left population are set as $\zeta_i^L = -\zeta_i^R$.

The tuning curves were chosen based on[75], where the position sensitivity of nucleus prepositus hypoglossi of the macaque were measured. Specifically we sampled the tuning curves randomly from

$$\zeta_i = \left( 0.032 \frac{\text{spks}}{\text{sec} \times \text{deg}^2} \right) E_T + 4.04 + \epsilon_i^\zeta \tag{27}$$

where the coefficients of the linear relationship are set based on[75], and $\epsilon_i^\zeta$ is a zero-mean, normally distributed random variable whose variance was set to obtain the same dispersion as experimentally observed in[75], and the measured correlation coefficient in[75] was 0.61. The weights $\eta_i$ were determined by an optimization process as in[31], ensuring that the system has an approximate continuum of fixed points in which the eye position variable $E_{OI}(t)$ spans the range of $-50°$ to $50°$ degrees. The precise values of $\zeta_i$, $\lambda_i^0$, and $\eta_i$ are listed in SI Data. 1.

*OMN activity and actual eye position.* The internal eye representation generates activity of the OMNs according to the following dynamics

$$\lambda_i^M(t) = \left[ k_i \left( E_{OI}(t) - E_{T_i} \right) \right]_+ \tag{28}$$

$$\tau_{MN} \frac{d}{dt} s_i^M = -s_i^M + \xi_i^M \tag{29}$$

where $\lambda_i^M$, the firing rate of OMN $i$, is assumed to be determined linearly and instantaneously from the synaptic readout $E_{OI}(t)$ of the oculomotor integrator network. We incorporate noise in activity of the OMNs by generating a stochastic spike train $\xi_i^M$ with rate $\lambda_i^M$ and with a CV of ~0.07 as described below. Each OMN generates a synaptic output $s_i^M$ which is determined by eq. (29).

Each OMN is assumed to innervate an extraocular muscle fiber and to contribute a signal $E_i$ to the actual eye position, which can be expressed as a convolution of $s_i^M$ with a double-exponential kernel or, equivalently, as the solution

of the differential equation

$$s_i^M = k_i \left( E_i - E_{T_i} \right) + r_i \dot{E}_i + m_i \ddot{E}_i \tag{30}$$

with parameters $k_i$, $r_i$, and $m_i$. Note that eqs. ((28)–(30)) are set up such that when the eyes are still $E_i = E_{OI}$ up to stochastic fluctuations ($s_i^M$ is equal at steady state to $\lambda_i^M$, up to stochastic fluctuations, due to eq. (29)).

Finally, the position of the eye is taken to be an average over all the OMN contributions:

$$E(t) = \frac{1}{N_m} \sum_{i=1}^{N_m} E_i(t) \tag{31}$$

When the eyes are still and at steady state the actual eye position $E$ matches the internal representation of the oculomotor integrator $E_{OI}$[76,77].

The parameters $k_i$, $r_i$, $m_i$, and $E_{T_i}$ of the OMNs were set as follows. First, the eye position threshold $E_T$ was sampled uniformly in the range $(-45°)–(-5°)$, with mean eye threshold of $-25°$, similar to[36], we discarded neurons with eye thresholds above $-5°$ since their contribution to eye position during straight ahead gaze is negligible. Second, we assumed an approximate linear relationship between $k$, $r$ and $E_T$, as observed in[37]. Thus, $k_i$ and $r_i$ were set as

$$k_i = \left( 0.18 \frac{\text{spks}}{\text{sec} \times \text{deg}^2} \right) E_T + 8.07 + \epsilon_i^k \tag{32}$$

$$r_i = \left( 0.02 \frac{\text{spks}}{\text{deg}^2} \right) E_T + 1.23 + \epsilon_i^r \tag{33}$$

where the coefficients of the linear relationships were set based on[37], and where $\epsilon_i^k$ and $\epsilon_i^r$ are zero-mean, normally distributed random variables whose variance was chosen to obtain the same dispersion as experimentally observed in[37] (correlation coefficients $R = 0.81$ for $k$ and $R = 0.67$ for $r$). In addition we imposed hard constraints $k \geq 1.1$, $r \geq 0.25$ according to[36]. The typical values for $k$, $r$ obtained from this procedure matched the values reported in[36]. For the acceleration coefficient, $m$, we randomly sampled values according to typical values reported in[36] while verifying that $m < r^2/(4k)$. The precise parameters values of $r$, $k$, and $m$ are given in SI Data. 2 and illustrated in Extended Data Fig. 9.

Our simulation included $N = 30,000$ neurons in the oculomotor integrator, $N_m = 1000$ OMNs, synaptic time constants of $\tau_s = 20$ ms, and visual feedback amplitude $A = 0.015$ (see below). We used a set of parameters that is biologically plausible, but note that other combinations of parameters can produce similar results. Specifically, several key parameters determine together the amplitude of the MSD curve: the number of neurons in the oculomotor integrator, their synaptic time constant, firing rate, non-linearity, and variability as stated in eq. (1)[33].

*Visual feedback.* We assume that the visual feedback during fixational drift is based on motion of the target on the retina, rather than an attempt to fix the absolute position of the target[24]. Therefore, the visual feedback signal $F(t)$ in eq. (23) is proportional to an estimate of the target angular velocity relative to the eye direction in the recent history. The estimate of velocity is generated using a signal that arrives to the oculomotor integrator with a delay $\tau_d \approx 70$ ms[9]:

$$F(t) = A \left[ E(t - \tau_d) - E_{OI}(t) \right] \tag{34}$$

where $E(t)$ is the actual gaze direction established as described above.

*Sub-Poisson spike trains.* Both oculomotor neurons and motoneurons are sub-Poisson with CV of the inter-spike interval <1. In order to take this into account in our model we used spike thinning by first generating Poisson spikes at a rate equal to the desired firing rate, multiplied by a factor $M$. From the resultant spike train we then used every $M$'th spike. This procedure keeps the average firing unchanged, but reduces the CV by a factor $1/\sqrt{M}$. When the firing rate is constant, this procedure is equivalent to sampling inter-spike intervals from a Gamma distribution. For oculomotor neurons we assumed CV ~ 0.22, for the OMNs we assumed the CV is linear with the mean inter-spike interval, (ISI) in similarity to the observed values in[48]

$$\text{CV} = 10^{-2} \times \left[ 6.34 + \left( 0.17 \frac{1}{s} \right) \text{ISI} + \epsilon_{CV} \right] \tag{35}$$

where the ISI is measured in ms, and $\epsilon_{CV}$ is a zero-mean, normally distributed random variable whose variance was chosen to obtain the same dispersion as experimentally observed in[48]. Note that these values are based on the empirical distribution of inter-spike intervals during fixational intervals and could therefore be influenced, in principle, by the small motion of the eye during fixational drift. However, we show in SI Notes (Contribution of oculomotor state to the CV of OMNs) that this influence is negligible. In addition we demonstrate that spiking noise affects the MSD curve only at relatively short time lags (up to ~10 ms, see Extended Data Fig. 10).

**Statistics.** No statistical method was used to predetermine sample size. Cells in which we were unable to consistently detect and isolate spikes due to low signal-to-noise ratio of the recordings were excluded from the analysis. In addition, trials in

which cells were not active for any fraction of the trial were excluded from the analysis. Error bars in Fig. 2e, f and in Extended Data Fig. 1 were calculated as described above (Standard error of the estimates for the covariance and R and Estimated contribution of central source to measured eye position).

**Reporting summary**. Further information on research design is available in the Nature Research Reporting Summary linked to this article.

## Data availability
Source data are provided with this paper. The raw data in this study, that includes recorded eye positions along OMN spike times have been deposited in the Zenodo database at 10.5281/zenodo.6076911. The data in this study was reported in published studies[38,63,64]. Source data are provided with this paper.

## Code availability
The code for the computational model is publicly available in github. (see https://github.com/The-Burak-lab/Fixational-drift-is-driven-by-diffusive-dynamics-in-central-neuralcircuitry)

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

## Acknowledgements

The authors would like to thank Gilad Ben-Shushan for providing Fig. 4a. This research was supported by the Israel Science Foundation grant No. 1733/13 and grant No. 1745/18, and in part by the Israel Science Foundation grant No. 1978/13. We acknowledge additional support from the Swartz foundation and from the Center for Brains, Minds and Machines (N.S.), the Israel Science Foundation grant No. 380/17 and the European Research Council grant No. 755745 (M.J.), and the Gatsby Charitable Foundation (Y.B.). This work was done in part while visiting the Simons Institute for the Theory of Computing (N.B.-S., N.S., and Y.B.). Y.B. is the incumbent of the William N. Skirball Chair in Neurophysics.

## Author contributions

The study was conceived by Y.B. with major inputs from all authors. The data was collected by M.J.; N.B-S. analyzed the neural recordings. The neural data was interpreted by N.B-S., M.J., and Y.B. with inputs from N.S. The theoretical model was developed by N.B-S., N.S., and Y.B. with inputs from M.J. Visualization of results was carried out by N.B.-S. and N.S. Numerical simulations were performed by N.B.-S. and N.S. Mean-squared displacement curves (experimental and theoretical) were analyzed by N.B.-S. and N.S. and interpreted by N.B-S., N.S., and Y.B. with inputs from M.J. The manuscript was written by N.B-S. and Y.B. with inputs from all authors. The manuscript was edited by all authors. The project was jointly supervised by M.J. and Y.B.; M.J. and Y.B. obtained funding.

## Competing interests

The authors declare no competing interests.
