## [Peer review file · Nature Communications]

REVIEWER COMMENTS

Reviewer #1 (Remarks to the Author):

In this study, the authors correlated oculomotor neuron (OMN) activity to subtle changes in fixational eye position (fixational drifts). They found that there was a correlation between OMN activity and drifts. However, the OMN activity was variable, such that predicting eye position from OMN activity resulted in much larger variance than is observed with real drifts. The authors argued that the variance between OMN's must have some shared component to it, consistent with the idea that a higher area (e.g. neural integrator) drives drifts. For this, they relied on modelling. They suggested that a network for implementing the neural integrator could drive the drift effects in OMN's.

Identifying neural drivers for drift eye movements is important. So, this aspect of the paper is relevant.

However, the paper is very hard to read. Some examples are provided below in the detailed comments, but this does not help the paper a great deal. It gets too technical, and even the technical parts need multiple re-reads to follow them, and the figure legends are not too helpful.

Also, I feel that the analysis of the neural activity seems too superficial. For example, what is the relationship between the correlation to drift and the tuning curve of each oculomotor neuron? Or between the correlation to drift and the position threshold of each oculomotor neuron? And, is the activity for neurons that are below threshold still correlated with drifts or not? Is the activity below thresholds really constant or is it variable? And is it more or less variable during drift than for neurons that are above the eye position threshold? Etc etc etc. It seems that the OMN data, which are the biggest pillar of the study, can be analyzed more richly than what was done, and this is really a missed opportunity.

Otherwise, relying on modelling, and saying that the neural integrator controls subtle eye position, is somewhat underwhelming. If drifts are affected by task demands, as with several citations in the paper, then it must be driven by the neural integrator (and even higher in the brain). So, there was a missed opportunity with this manuscript to provide much more than the simple intuition that drift is centrally driven, which could have been made based on past papers. This could have been done with much more detailed analyses of the OMN data. For example, what would be the contributions of INN's to movements of the other eye, and would this explain the differences in drifts between the two eyes during simple fixation? Answering a question like this would be extremely interesting.

Another example of the above ideas is the kernel computation in Fig. 2a. What was the distribution of kernel peak values across neurons? The paper could really use some deep dive in the neural analyses (along with more details of eye movement behavior, like binocular coordination).

Also, is the prediction of eye position in Fig. 2c better or worse during the pursuit phase with larger eye movements? This should be informative for the presented mechanisms.

Some more specific comments below:

- line 6: the authors state that they “identify its origin in central neural circuitry”. This sentence seems to be quite vague for me, and potentially misleading. For example, oculomotor neurons are pretty peripheral in the brain. More importantly, identifying correlations between oculomotor neurons and drifts does not necessarily mean that the oculomotor neurons or the neural integrator are the “origin” of the phenomenon. These neurons could reflect more upstream modulations that then get implemented by the oculomotor neurons. The implementation by the neural integrator and oculomotor neurons is inevitable if the drifts are centrally controlled.

- it’s also a bit unusual for this journal to have citations in the abstract, no?

- line 37: this citation is very odd for such a general sentence on the similarity between monkey and human oculomotor systems

- Fig. 1a: explain how smoothing was done in the legend

- Fig. 1b: explain how “predicted” firing rate was obtained

- paragraph of line 62: it is not clear whether you did the correlation analysis on the whole trial (i.e. including pursuit) or only on the fixation period. It should be only on the fixation period for the purpose of this paper. Please clarify this and explicitly mention it

- Fig. 2b,e, put a bounding box around the legend in the figure. I am having a hard time identifying which blue and red dots are data points and which are part of the explanatory legend

- Fig. 2e: this figure is very hard to understand. How is a fraction larger than 1 (x-axis)? Or is it a percentage? And, what is the unit of the y-axis? Is it deg? The authors are leaving way too much for the reader to interpret, which renders the figure (and paper) very hard to follow. The main text doesn't help either. For example, I still do not understand why the position of the black diamond allows concluding that a "large fraction" in the variance is due to a central source. Even the greek symbol in the x-axis is not explained adequately.

- line 89: too much mathematical jargon. The authors need to explain what is R^2 and why is the relationship between $\langle R^2 \rangle$ and $\langle R \rangle^2$ is like they say, etc etc etc. The paper is quite hard to read.

Reviewer #2 (Remarks to the Author):

The authors present new analyses of previously published recordings from two rhesus monkeys, indicating that fixational drift is correlated with neural activity, and that this activity originates in central neural circuitry within the oculomotor system. Specifically, the data suggest that most of the fixational drift variance arises upstream of the ocular motoneurons. I find these conclusions rather convincing, though the correlations are somewhat weak. Overall, this is high-quality research that will be of interest to the oculomotor neuroscience field.

General Comments:

- a. The use of only two monkeys may limit the reproducibility of the results. However, the data from both monkeys appears similar, so I have no strong reason to suspect that these results would not replicate in data from additional animals.
- b. Methodological details concerning neuronal and search coil recordings should be provided in the present MS (rather than referring the reader to prior publications), as they might be important for data interpretation.

Minor comments:

a. If possible, the authors should provide the source code for the computational model, to help others replicate their work.

b. Extended Data Fig. 7: The word “Shuffled” should be “Shuffled”

c. I found a few omissions in the literature review, detailed below:

- Line 18. “...specific areas of interest.” Also Otero-Millan et al, PNAS 2013
- Line 21. “...by microsaccades.” Also Martinez-Conde & Macknik, Philos Trans R Soc Lond B Biol Sci. 2017
- Lines 24-28, on the functional roles of drift. Here, it would be appropriate to indicate that both drifts and microsaccades help prevent perceptual fading during fixation (McCamy et al, J Physiol. 2014), though only microsaccades can restore visibility after fading has occurred (McCamy et al, J Neurosci 2012).
- Line 37. “...highly similar to that of humans.” Ref. #4 is not appropriate at the end of this sentence. Here the authors might refer to some of the pioneer neurophysiological studies of eye movements in the primate, such as those by Wurtz in the late 1960s.
- Line 219. “...to saccades.” Also McCamy et al, J. Physiol. 2014, with the limitations noted above.
- Line 217. Drift is also influence by certain non-visual task parameters, such as fatigue. Specifically, increased time-on-task results in increased drift velocities (Di Stasi et al, Eur. J. Neurosci. 2013; Di Stasi et al, Exp. Brain Res. 2015).

Reviewer #3 (Remarks to the Author):

Summary:

This study uses motor neuron recordings and eye movements to evaluate the origin and scale of motor noise in the oculomotor system. Specifically, the work presents evidence that fixational eye position drift is consistent with a (super) diffusion model based on added noise upstream from oculomotor nuclei. OMN units are known to be quite precise (very low Fano factors), but this work explores the behavioral implications for the small amounts of observed variability considered at the population level. A clever feature of the analysis is to derive a linear model of each unit’s contribution to large scale eye movements and then apply that model to fixation. The data are previously recorded OMN unit responses and eye movement behavior in monkeys. Linear analysis is used to estimate the fraction of horizontal eye position variance arising upstream from abducens. A spiking model of an upstream integrator and simulated motor neuron activity are shown to replicate the mean squared drift distance vs. time relationship observed in the behavioral data.

General comments:

Generally, I found the subject to be interesting and the presentation to be good, but so terse that it does not make explicit how this work expands our understanding of oculomotor control.

If the strength of the paper is a spiking model of the neural integrator, highlighting what this model predicts that the rate-based model did not would be helpful. If the strength of the paper is identifying the origin of variability in fixational eye position, then a more thorough analysis of the behavioral data would be helpful. If the goal is to analyze the motor unit population, then better justification for treating the units as independent and not using the recorded spikes would help. Much of the presentation relies on the model's agreement with the mean squared horizontal eye displacement (MSD) as a function of time interval, but a substantial range of the experimental data appears to be dominated by measurement noise. This makes it difficult to judge the significance of the differences in the log MSD vs log t curves for the models in the extended figures.

Specific Comments:

One of my questions is the contribution of near measurement threshold signals to the data curve used to motivate the success of the model. The task design for the original experiment necessitated the use of short time windows where the amount of drift is quite small and measurement noise is an issue. If I interpret Fig 3 and EFig 8 correctly, nearly half the data range on the log MSD vs log t plot is dominated by measurement noise. I think that the behavior-model comparisons would be on more solid footing if the subtraction (or filtering) of measurement noise occurred before the calculation of MSD. The reader needs to know more about the statistics of measurement noise vs. signal, the sampling rate, and the distribution of inter-saccade intervals that were analyzed to evaluate the figure. At the very least, EF8 needs to be incorporated into Figure 3. This problem makes it difficult to judge the significance of the small differences in log MSD vs log t plots for different models in the extended figures.

Line 52: citations of the literature being referred to would be helpful.

Line 120: If the noise correlations between OMNs has been measured, please cite. If not, what is the basis for the assumption of independence?

Line 122 Does MSD refer only to horizontal displacement?

Fig 2e More explanation of the axes is needed – is this a log scale?

(Note, use of two differently typeset epsilons to indicate distinct quantities makes it hard to refer to)

Line 315: cite evidence that OMNs have negligible noise correlations to support this assumption

Line 374: Were no primate data available to estimate the tuning functions? What is the rationale for the range chosen?

Line 425 unneeded comma “where the ISI, is measured”

EFig 9: How sensitive is the model to the distribution of integrator tuning functions? Is there any data in the primate literature to support the choice made?

Sep, 2021

Response for the reviewers

We would like to thank the reviewers for their insightful comments. We addressed all the comments (see below), and we believe that this has led to a significant improvement of the manuscript. Some of the reviewer comments identified points in the manuscript which were not explained in sufficient detail. One of the reasons for this is that the manuscript was transferred without any modification from another journal (*Nature*). The formatting guidelines of *Nature Communications* are more flexible, allowing for a longer article with more figures. Therefore, we substantially revised the manuscript in the resubmission in order to expand on topics that were previously discussed more briefly within the main text. Alongside the expanded text, the manuscript now includes five figures instead of four, with several new panels (2e, 3a, 4a-c, 5e-f), and new supplementary figures (2a - inset, 2b, 4, 7).

Please see below our detailed responses to the reviewer comments. We have highlighted changes in the manuscript in yellow.

Reviewer 1:

In this study, the authors correlated oculomotor neuron (OMN) activity to subtle changes in fixational eye position (fixational drifts). They found that there was a correlation between OMN activity and drifts. However, the OMN activity was variable, such that predicting eye position from OMN activity resulted in much larger variance than is observed with real drifts. The authors argued that the variance between OMN's must have some shared component to it, consistent with the idea that a higher area (e.g. neural integrator) drives drifts. For this, they relied on modelling. They suggested that a network for implementing the neural integrator could drive the drift effects in OMN's.

Identifying neural drivers for drift eye movements is important. So, this aspect of the paper is relevant.

However, the paper is very hard to read. Some examples are provided below in the detailed comments, but this does not help the paper a great deal. It gets too technical, and even the technical parts need multiple re-reads to follow them, and the figure legends are not too helpful.

We have expanded the main text, which was previously shorter since the manuscript was transferred from a journal with tight restrictions on length (among them, restrictions on the figure captions). We hope that these changes have significantly improved the readability.

Also, I feel that the analysis of neural activity seems too superficial. For example, what is the relationship between the correlation to drift and the tuning curve of each oculomotor neuron? Or between the correlation to drift and the position threshold of each oculomotor neuron? And, is the activity for neurons that are below threshold still correlated with drifts or not? Is the activity below thresholds really constant or is it variable? And is it more or less variable during drift than for neurons that are above the eye position threshold? Etc etc etc. It seems that the OMN data, which are the biggest pillar of the study, can be analyzed more richly than what was done, and this is really a missed opportunity.

Single OMN properties have been extensively studied in the context of large eye motions, and it would indeed be interesting to examine how these properties relate to the correlation between single OMN activity and fixational drift.

Before we explain the analyses that we performed in response to this comment, it is important to note that extracting such relationships is highly challenging: fixational eye motion is tiny, and the variation in OMN activity, related to fixational motion, is small compared to the intrinsic neural variability. For this reason, our main analysis in the manuscript required statistics from dozens of neurons - in order to reach a reliable conclusion about the mere existence of substantial correlation between OMN activity and eye motion during fixation. Any attempt to analyze this correlation in relation to a third factor (e.g. properties of individual neurons), is more subtle, and would require extremely large datasets.

We analyzed the OMN data in more detail according to the specific suggestions made above in the reviewer comment. We provide as an appendix to this letter figures showing the results of this analysis: specifically, we looked for a relation between the correlation coefficient and the slope of the tuning curve k (Revision Figure 1a), the threshold (1b), the velocity sensitivity (1c). We also examined the relationship between the inferred shared variance coefficient χ and the same quantities (panels d-f). Overall, we didn't identify significant statistical relations between the parameters of the OMN and our variables of interest, i.e. the correlation, during fixation, between the predicted eye motion from a single OMN to the actual eye motion, and the fraction of variance in actual eye position that can be explained from a single OMN (Revision Figure 1).

In addition, we compared the data against similar analysis performed on our simulated data (also shown in Revision Figure 1 panels a-f, to the right of the analysis of experimental data). In the simulations we used the same OMN tuning curves as we measured from the data and over a similar number of trials (~100 trials per cell). Results from simulations are extremely noisy, roughly in a similar manner as in the data – demonstrating the difficulty in collecting reliable statistics on these features.

Regarding the question on correlation to motion of neurons below threshold: obviously, neurons that are completely below threshold do not discharge action potentials and therefore their activity cannot be correlated with the motion. Nonetheless, one can still ask whether OMNs that are close to the threshold are still correlated with the eye position. Revision Fig. 1g shows the inferred correlation coefficient as a function of the firing rate at the fixation point: the data is very noisy, but does not indicate a reduction of correlation in cells with low firing rates. Note that as mentioned above, the estimate for the correlation coefficient in this figure is more noisy than in other panels since we could only average over trials at similar fixation positions.

In panels h,i,j we look for additional correlations between our variables of interest and the mean discharge rate or CV of a cell during a trial. Similarly to panel g, in these panels we had to distinguish between trials which started at different fixation positions, since the rate and CV depends on the position of the eye. Therefore, each point in these panels corresponds to a cell at a specific fixation location. Similar analysis was performed in the simulation, i.e. we simulated the same OMNs at different fixation positions. At all panels, the Spearman correlation coefficient and its p-value are reported, and error bars represent the standard error of the mean.

Overall, the analysis shown in Revision Figure 1 points to the difficulty of identifying subtle correlations between single OMN properties and their correlation to fixational drift. Hence, we decided not to include this figure in the manuscript. However, we will be happy to reconsider this choice if Reviewer 1 believes that including this analysis will enhance the paper.

Otherwise, relying on modelling, and saying that the neural integrator controls subtle eye position, is somewhat underwhelming. If drifts are affected by task demands, as with several citations in the paper, then it must be driven by the neural integrator (and even higher in the brain). So, there was a missed opportunity with this manuscript to provide

much more than the simple intuition that drift is centrally driven, which could have been made based on past papers.

Our findings are highly significant for several reasons:

- I. Even though it may be inferred that neural activity upstream of the ocular motoneurons is correlated with fixational drift, this has not been shown previously.
- II. The main source of the motion is unknown, and it has been hypothesized that much of the motion might arise peripherally, in the dynamics of the ocular motoneurons or the muscles. The fact that fixational drift is modulated by task demands or by the stimulus indicates (as noted by Reviewer 1) that this motion is influenced by various brain areas within the visual-motor loop, but it does not imply that most of the variance in the motion is centrally driven. Note also that fixational drift is observed even in complete darkness, which indicates that the drive for the motion does not require the presence of any stimulus. Whether or not most of the drive for the motion is upstream of the OMNs was not known previously.
- III. We offer a specific mechanism that might serve as the main drive of the motion: random noise that drives diffusive drift within the oculomotor integrator. The plausibility of this hypothesis is backed up by detailed quantitative modelling of the dynamics, in which the key parameters are consistent with those of the primate oculomotor system. The fact that the model explains well the magnitude and the detailed structure of the empirical MSD curves is highly significant. The fact that visual feedback modulates the MSD curves is consistent with the observation (cited extensively in our manuscript) that the stimulus can influence the detailed statistics of the motion.

Taken together, our results offer a new perspective on the origin of fixational drift. We revised the introduction to bring up these points more clearly.

This could have been done with much more detailed analyses of the OMN data. For example, what would be the contributions of INN's to movements of the other eye, and would this explain the differences in drifts between the two eyes during simple fixation? Answering a question like this would be extremely interesting.

Indeed this is a very interesting question, unfortunately, we don't have simultaneous recording of both eyes during the primate task. Our theory supports the idea that to some extent, correlation in fixational drift between the two eyes should be present, yet these correlations might be only partial, due to the bilateral organization of the system. This is the topic of a separate study that we are currently pursuing, in collaboration with a neurophysiology lab working with zebrafish and goldfish.

Another example of the above ideas is the kernel computation in Fig. 2a. What was the distribution of kernel peak values across neurons? The paper could really use some deep

dive in the neural analyses (along with more details of eye movement behavior, like binocular coordination).

As can be seen in the Revision Figure 2, the majority of OMNs have a kernel peak value of ~1-3 deg/spk. In addition we couldn't find significant correlation to the correlation with drift for example. We don't have binocular data.

Also, is the prediction of eye position in Fig. 2c better or worse during the pursuit phase with larger eye movements? This should be informative for the presented mechanisms.

The prediction of eye position during larger eye movements is better, compared with the prediction during fixation. The reason is that the signal is stronger during the large eye movements. This was previously demonstrated in *M. Joshua, S.G Lisberger, 2014, JNP.*

- line 6: the authors state that they "identify its origin in central neural circuitry". This sentence seems to be quite vague for me, and potentially misleading. For example, oculomotor neurons are pretty peripheral in the brain.

We have now corrected the phrase to "identify its origin in central neural circuitry within the oculomotor system, upstream to the ocular motoneurons (OMNs)." (line 5-6). Note that our methodology is based on identifying correlations between the activity of individual OMNs and the motion, but much of the analysis and argumentation is devoted to showing that these correlations are a signature of a common upstream input to the OMNs. While the OMNs are often considered to be quite peripheral, we believe that it's appropriate to refer to the oculomotor integrator as central.

More importantly, identifying correlations between oculomotor neurons and drifts does not necessarily mean that the oculomotor neurons or the neural integrator are the "origin" of the phenomenon. These neurons could reflect more upstream modulations that then get implemented by the oculomotor neurons. The implementation by the neural integrator and oculomotor neurons is inevitable if the drifts are centrally controlled.

This point is correct. Showing that the drive for the motion is upstream of the OMNs does not necessarily imply that it is driven by the internal dynamics of the neural integrator. Our proposal, that diffusive drift arises primarily from noise-driven diffusion within the oculomotor integrator is based on extensive and detailed theoretical modeling, which demonstrates that:

- (i) The observed MSD curves of fixational drift are indicative of diffusive motion in the output of the oculomotor integrator with statistics that are, to a good approximation, a simple random walk.
- (ii) Introducing noise into the dynamics of oculomotor integrator neurons, with parameters that are based on known features of the circuit, yields diffusive dynamics in the output of the integrator, with the correct amplitude.

We believe that these results make a very strong argument in support of the hypothesis. The first result of our manuscript (that fixational drift is driven upstream of the OMNs) is based on direct analysis of the neural recordings, whereas the second result is based on theoretical argumentation. The second part of the work provides a new perspective on the possible origin of the motion, which is strongly backed by theoretical arguments.

There are, however, alternatives: noisy inputs to the integrator might drive diffusive motion along the attractor, instead of the intrinsic noise of the oculomotor integrator neurons, as we pointed out in the discussion. In order for noisy inputs to drive most of the diffusive dynamics inside the integrator, and to account for most of the variance, the contribution of the intrinsic noise within the integrator network should be decreased substantially, potentially requiring unlikely parameters. For example, to decrease the amplitude of the motion by one order of magnitude a ten-fold increase in the number of oculomotor neurons is required, which would bring the number of oculomotor integrator neurons to be of order 100,000 neurons, and this seems less likely. We now discuss in the main text the dependence of the diffusion coefficient of the motion along the integrator attractor on the network parameters.

Another possible alternative is that the integrator is fed by a signal which has characteristics of a simple diffusive process. While it is natural for the oculomotor integrator to generate such statistics (as this is exactly the expected behavior of a line attractor fed by temporally uncorrelated noise), it is less clear why upstream circuits in the oculomotor pathway would generate this type of motion.

- It's also a bit unusual for this journal to have citations in the abstract, no?

Thanks, this is correct. The manuscript was transferred from another journal. We removed the citations from the abstract.

- line 37: this citation is very odd for such a general sentence on the similarity between monkey and human oculomotor systems

We cited this particular paper (*Snodderly et al., 2016, Vision Res.*) because it highlights the similarity between monkey and human eye motion specifically during fixation. We agree that the statement made in line 37 of the original submission was more general and therefore the lack of an earlier reference could seem odd. We now cite an earlier work (*Fuchs, 1967, J. Physiology*) for the general statement, and cite *Snodderly et al., 2016, Vision Res.* specifically in the context of fixation (now at line 44).

- Fig. 1a: explain how smoothing was done in the legend

Done, changes are highlighted in Fig 1 caption.

- Fig. 1b: explain how "predicted" firing rate was obtained

We now briefly describe this in the figure caption and refer to the methods section where additional details are available.

- paragraph of line 62: it is not clear whether you did the correlation analysis on the whole trial (i.e. including pursuit) or only on the fixation period. It should be only on the fixation period for the purpose of this paper. Please clarify this and explicitly mention it.

Thanks for this comment. Of course, the analysis was performed on the fixation segments alone, and we edited the text in order to make this much more clear (see, for example, line 76).

- Fig. 2b,e, put a bounding box around the legend in the figure. I am having a hard time identifying which blue and red dots are data points and which are part of the explanatory legend

Done.

- Fig. 2e: this figure is very hard to understand. How is a fraction larger than 1 (x-axis)? Or is it a percentage? And, what is the unit of the y-axis? Is it deg? The authors are leaving way too much for the reader to interpret, which renders the figure (and paper) very hard to follow. The main text doesn't help either. For example, I still do not understand why the position of the black diamond allows concluding that a "large fraction" in the variance is due to a central source. Even the greek symbol in the x-axis is not explained adequately.

Thanks for the comment, in order to clarify the main message, that a "large fraction" of the variability in the final eye position is due to a central source, we took the following steps:

- (1) We further elaborate in the main text on the variable which is shown in the figure, i.e. the greek letter in the x-axis, which is denoted in the revised manuscript by χ (instead of ε in the original submission). In brief, this is indeed an estimate of the fraction of variability generated by the central source. The actual fraction should be smaller than 1, but there is nothing that prevents individual estimates from exceeding this value (as explained now in lines 137-138 and 141-143 and corroborated by the new simulations, shown in Fig. 5e and EFig 4 - see point 5 below). The estimates are based on the empirical covariance between the actual eye motion and the predictor obtained from the single OMN activity, and are very noisy. In other words, the value of χ obtained from individual sessions is a noisy random variable, mostly due to the single OMN noise. To obtain an informative estimate for χ we pool over the estimates from the individual neurons.
- (2) One of the reasons why Fig. 2e (Fig. 2f in the resubmitted manuscript) was difficult to digest, is that it shows on the x-axis the values of χ , and in the y-axis our estimate of how accurately we estimate them (our confidence interval for each measurement, which varies from cell to cell due to variation in various parameters such as the number of trials, the firing rate and its sensitivity to eye position, etc.). We now show, first, in panel 2e, a more conventional figure where the estimates χ are shown sequentially from all trials in one

monkey, ordered by their confidence interval. Here the y axis simply represents χ , and the confidence interval is represented by a vertical line.

Note, in both panels, that in those cases where the estimate of χ was outside of the expected range $[0, 1]$, the estimate of the error was large (the confidence interval was wide). Therefore, these measurements are still consistent with a value which is within the expected range, and the weighted mean (see below) is only weakly influenced by these measurements.

- (3) We added text to the y-axis of Fig. 2f (previously Fig. 2e): *Estimated error of χ* . Therefore both axes are dimensionless.
- (4) We further elaborate in the main text on the conclusion from this figure. In brief, since we have multiple estimates of χ , the fraction of variability in final eye position, contributed from a central source, we can average across all of them in order to obtain a better estimate of the mean value. Since each of these estimates is noisy by itself, and some are significantly more noisy than others, the appropriate way to do this is to calculate a weighted mean, which takes into account the confidence interval for each individual estimate.

This weighted mean is represented by the black diamond in Fig. 2f (previously Fig. 2e). The position of the black diamond along the x axis is the weighted mean itself (~ 0.8), and it's position along the y axis represents our confidence interval for the weighted mean, which is much smaller than the confidence interval for individual OMNs because it is based on all the sessions.

The fact that the weighted mean turns out to be close to one, and is significantly different from zero (due to the relatively low standard error of the mean we calculated - roughly 10% of the mean value), is the basis of our conclusion that most of the variability in measured eye position, at a 350ms time lag, can be attributed to a source upstream to the OMN.

- (5) We performed similar analysis on simulated data from our model, which produced similar results as obtained from the real data. In the analysis of the simulated data we used a similar number of trials per cell as in the measured data. The results are shown in a new figure panel, Fig. 4e. The similarity to the results from the experimental data is in two aspects: first, the analysis produces similar estimates of the mean values and their error. Second, the spread of estimates for χ across different OMNs is similar. In particular, note that, in similarity to our results from the experimental data, when we applied the same analysis to the simulated data we got, for some of the OMNs, an estimate larger than 1 for χ (the measured fraction of variability).

- line 89: too much mathematical jargon. The authors need to explain what is R^2 and why is the relationship between $\langle R^2 \rangle$ and $\langle R \rangle^2$ is like they say, etc etc etc. The paper is quite hard to read.

Thanks for this comment. We now explain the notation explicitly (the use of angular brackets to represent a mean is common in physics but less so in other disciplines). We have also edited and expanded the section on the central source, which was indeed terse in the original submission, and we believe that it is now easier to follow this section.

Reviewer 2

The authors present new analyses of previously published recordings from two rhesus monkeys, indicating that fixational drift is correlated with neural activity, and that this activity originates in central neural circuitry within the oculomotor system. Specifically, the data suggest that most of the fixational drift variance arises upstream of the ocular motoneurons. I find these conclusions rather convincing, though the correlations are somewhat weak. Overall, this is high-quality research that will be of interest to the oculomotor neuroscience field.

General Comments:

a. The use of only two monkeys may limit the reproducibility of the results. However, the data from both monkeys appears similar, so I have no strong reason to suspect that these results would not replicate in data from additional animals.

We agree.

b. Methodological details concerning neuronal and search coil recordings should be provided in the present MS (rather than referring the reader to prior publications), as they might be important for data interpretation.

Additional details regarding the experimental settings and methodology are now included in the methods section.

Minor Comments:

If possible, the authors should provide the source code for the computational model, to help others replicate their work.

Thanks for this suggestion. We prepared annotated code and placed it on GitHub (currently private). We provide the link to the code in the revised manuscript, and will make it public upon publication.

b. Extended Data Fig. 7: The word “Shuffeled” should be “Shuffled”

Thanks, we fixed this.

I found a few omissions in the literature review, detailed below:

• Line 18. “...specific areas of interest.” Also Otero-Millan et al, PNAS 2013

Added.

• Line 21. "...by microsaccades." Also Martinez-Conde & Macknik, *Philos Trans R Soc Lond B Biol Sci.* 2017

Added

• Lines 24-28, on the functional roles of drift. Here, it would be appropriate to indicate that both drifts and microsaccades help prevent perceptual fading during fixation (McCamy et al, *J Physiol.* 2014), though only microsaccades can restore visibility after fading has occurred (McCamy et al, *J Neurosci* 2012).

Added.

• Line 37. "...highly similar to that of humans." Ref. #4 is not appropriate at the end of this sentence. Here the authors might refer to some of the pioneer neurophysiological studies of eye movements in the primate, such as those by Wurtz in the late 1960s.

We cited this particular paper (Snodderly et al., 2016, *Vision Res.*) because it highlights the similarity between monkey and human eye motion specifically during fixation. We agree that the statement made in line 37 of the original submission was more general and therefore the lack of an earlier reference could seem odd. We now cite an earlier work (Fuchs, 1967, *J. Physiology*) for the general statement, and cite Snodderly et al., 2016, *Vision Res.* specifically in the context of fixation.

• Line 219. "...to saccades." Also McCamy et al, *J. Physiol.* 2014, with the limitations noted above.

Added.

• Line 217. Drift is also influenced by certain non-visual task parameters, such as fatigue. Specifically, increased time-on-task results in increased drift velocities (Di Stasi et al, *Eur. J. Neurosci.* 2013; Di Stasi et al, *Exp. Brain Res.* 2015).

Added.

Reviewer 3

Summary:

This study uses motor neuron recordings and eye movements to evaluate the origin and scale of motor noise in the oculomotor system. Specifically, the work presents evidence that fixational eye position drift is consistent with a (super) diffusion model based on added noise upstream from oculomotor nuclei. OMN units are known to be quite precise (very low Fano factors), but this work explores the behavioral implications for the small amounts of observed variability considered at the population level. A clever feature of the analysis is to derive a linear model of each unit's contribution to large scale eye

movements and then apply that model to fixation. The data are previously recorded OMN unit responses and eye movement behavior in monkeys. Linear analysis is used to estimate the fraction of horizontal eye position variance arising upstream from abducens. A spiking model of an upstream integrator and simulated motor neuron activity are shown to replicate the mean squared drift distance vs. time relationship observed in the behavioral data.

General comments:

Generally, I found the subject to be interesting and the presentation to be good, but so terse that it does not make explicit how this work expands our understanding of oculomotor control.

We thank Reviewer 3 for pointing out that the presentation in the original submission is good but terse. Terseness is partly due to the formatting requirements that applied in our original submission (the manuscript was transferred to *Nature Communications* without modifications from another journal, following the recommendation of the editors). We have now revised the manuscript to include a more detailed exposition of the goals and implications of the work - specifically regarding the second part of the work, in which we propose a likely origin for the motion, relying mostly on theoretical arguments and the empirical statistics of the motion.

If the strength of the paper is a spiking model of the neural integrator, highlighting what this model predicts that the rate-based model did not would be helpful. If the strength of the paper is identifying the origin of variability in fixational eye position, then a more thorough analysis of the behavioral data would be helpful. If the goal is to analyze the motor unit population, then better justification for treating the units as independent and not using the recorded spikes would help.

In our view, the paper reports on two main results, which complement each other:

- (i) We show that most of the motion variance during fixational drift is driven upstream of the OMNs. It was previously possible to deduce indirectly that some of the variance is driven centrally, because of the modulation of motion statistics by the task or stimulus - but this doesn't imply that most of the motion is driven centrally. Here we show this much more directly, by analysing neural activity.
- (ii) We propose a completely new mechanism for the motion: diffusive drift in the output of the oculomotor integrator. Here we base our argumentation on theoretical modelling of dynamics in the oculomotor neural circuit, combined with analysis of the motion itself. The mechanism naturally explains both the magnitude and the detailed statistics of the motion, and is therefore, in our opinion, highly compelling.

We note that a large number of works examined the MSD curves of fixational drift, yet we are not aware of any work that proposed a mechanism that explains these curves, in a

model that is grounded in the physiology of the oculomotor system and is formulated at the level of a detailed theory of neural network dynamics.

We edited the introduction accordingly.

We cannot use the recorded OMN spikes in order to confirm our assumption that noise in these neurons is independent, because there are no simultaneous recordings from more than one OMN. Even if such recordings were available at a sufficiently large scale to enable meaningful statistical analysis, it is not clear that it would be possible to directly distinguish between shared variability which is coming from an upstream source, and shared variability that is generated internally within the abducens. We base our assumption of independence of noise in different OMNs on several experimental studies (see below, in our response to the detailed comments).

In the main text of our original submission we didn't explicitly mention that the oculomotor integrator network model was adapted from a rate model to a spiking model. This is a very important point. We mention this explicitly in the main text (lines 212-213).

Regarding a more thorough analysis of the data: our success in identifying a correspondence between fixational drift and single OMN activity is highly non-trivial, given the fact that noise dominates the estimates of eye position generated by the spiking activity of single OMNs (as compared with the small amplitude of the actual motion). Thus, the analysis required pooling over dozens of cells, with hundreds of trials for each cell. Previous works examined how specific properties of OMNs correlated with their contribution to eye motion, in the context of large eye motions. However, we are not convinced that, with our dataset, it is possible to extract meaningful statistics on similar features during fixational drift (see Revision Figure 1).

Much of the presentation relies on the model's agreement with the mean squared horizontal eye displacement (MSD) as a function of time interval, but a substantial range of the experimental data appears to be dominated by measurement noise. This makes it difficult to judge the significance of the differences in the log MSD vs log t curves for the models in the extended figures.

There are two types of noise which are affecting our analysis. The first one is the intrinsic variability of OMNs. During fixational drift this noise is dominating the predictors of eye position generated from the spiking activity of individual OMNs, and this is a central challenge that we had to confront in order to produce meaningful conclusions from the data. Our success in doing so relied on the availability of data from several dozen OMNs.

A second source of noise (mentioned in the comment above) is the measurement noise coming from the eye tracking system. This noise is significantly affecting the MSD curves generated from the raw measurements up to time scales of order 0.1s. We are confident that we can subtract this contribution to the MSD, to obtain reliable measurements of the actual eye motion's MSD. We added text in the methods section to support our noise subtraction methodology. In addition we added panels to EFig 8 which support our methodology and provide evidence that

our results lie on solid ground. We provide additional details on this question in response to a specific comment below.

Specific comments:

One of my questions is the contribution of near measurement threshold signals to the data curve used to motivate the success of the model. The task design for the original experiment necessitated the use of short time windows where the amount of drift is quite small and measurement noise is an issue. If I interpret Fig 3 and EFig 8 correctly, nearly half the data range on the log MSD vs log t plot is dominated by measurement noise. I think that the behavior-model comparisons would be on more solid footing if the subtraction (or filtering) of measurement noise occurred before the calculation of MSD. The reader needs to know more about the statistics of measurement noise vs. signal, the sampling rate, and the distribution of inter-saccade intervals that were analyzed to evaluate the figure. At the very least, EF8 needs to be incorporated into Figure 3. This problem makes it difficult to judge the significance of the small differences in log MSD vs log t plots for different models in the extended figures.

Note that, unfortunately, it is impossible to subtract measurement noise on a trial-by-trial basis before the calculation of the MSD, since the noise (by nature) is unknown to us. Our single trial measurements can be thought of as including two contributions: a signal (the actual eye position), and measurement noise. Attempting to filter out the measurement noise in individual trials using temporal smoothing would necessarily also apply similar smoothing to the signal, and this would distort the shape of the measured MSD curve, which is crucial to our analysis. However, we can reliably estimate the contribution of the noise to the MSD and subtract it (see below). This allows us to obtain reliable estimates of the eye motion MSD even at relatively short time lags, for which an alternative approach (based on temporal smoothing of single trials) would substantially distort the measurement. In order to justify the methodology used to subtract measurement noise and to strengthen the significance of the results we took the following steps:

- (1) We highlighted in the text that actually about 15% of the data points that we plot in Fig.3 are below the measurement noise variance. In addition we added to Fig.3 a panel which presents the MSD vs. time lag using linear axes, in which it is easier to appreciate the fraction of time lags which reside below the noise level.
- (2) We added text in the methods section, showing mathematically why subtracting the measurement noise variance from the MSD is justified. In brief, the point which one should remember is that the MSD is an average quantity, and as long as the variance of the measurement noise can be estimated precisely, it is meaningful to subtract it from the variance of the measured displacement, even when the two are of comparable size.

For example, if our estimate of the measurement noise contribution to the MSD is $10^{-3} \pm 10^{-4} \text{ deg}^2$, then the approach will start to break down when the actual MSD of the motion is comparable to 10^{-4} deg^2 (or smaller), not to 10^{-3} deg^2 . Indeed, when looking at

the subtracted MSD (EFig. 2a) at very short time scales (up to $\sim 2 \times 10^{-2}$ s), we observe that the results become more erratic and less reliable than at long time scales - which is why we plot the MSD curve in the main paper only for time scales exceeding $\sim 2 \times 10^{-2}$ s. Our ability to reliably recover the underlying MSD is evident also in simulations, in which we add noise to individual trials and then subtract the noise variance from the MSD curves - see below.

- (3) We verified the reliability of the measurement noise removal methodology via simulations in the following manner: we added simulated measurement noise to the simulated eye trajectories and then calculated and plotted the MSD. Subsequently, we subtracted the measurement noise variance from this calculated MSD and plotted it. Finally we plotted the ground truth MSD, i.e. the MSD calculated from the simulated eye trajectories without measurement noise. These panels were incorporated into the new EFig 2b, where it is possible to appreciate that the simulated curves are very similar to the MSD curves which were calculated from the experimental data.
- (4) We added EFig 4, showing the distribution of inter-saccade intervals in our data set, and a detailed explanation in the Methods section of how the MSD curves were evaluated from fixational segments of variable length. In this section we mention the sampling rate (mentioned also in Methods), and the estimated variance of the measurement noise.

Line 52: citations of the literature being referred to would be helpful

We added relevant citations (now in line 60).

Line 120: If the noise correlations between OMNs has been measured, please cite. If not, what is the basis for the assumption of independence?

As noted in this comment, we estimate the potential contribution of spiking variability of OMNs to the motion under an assumption that this variability generated within the abducens is independent in different OMNs. This assumption is primarily based on anatomical tracing, that did not find axon collaterals within the abducens nucleus (Baker, R and McCrea, R. The parabducens nucleus, 1979). We now mention this explicitly in the discussion section and cite the relevant references there. In lines 171-172 we refer to the discussion section

This question is important not only in the context of the MSD curves and our theoretical model, but also for the interpretation of the correlation between OMN activity in the abducens and fixational eye motions in the first part of the paper: specifically, for reaching the conclusion that the results point to a source upstream of the abducens. Therefore, we mention this point and refer to the new paragraph of the discussion section also in lines 110-111.

In the context of the MSD curves it is important to note that abducens OMNs (from which we recorded) are not the sole contributors to the motion. Following the study mentioned above, other studies (Evinger, Baker, McCrea, *Brain research*, 1979 and 1982) found that there are

axon collaterals in the medial rectus nucleus (the medial rectus muscle is antagonistic to the lateral rectus muscle which is innervated by OMNs in the abducens). One manuscript (cited as Ref. 53 in our revised submission) explicitly discusses the difference, in this respect, between the abducens nucleus and the medial rectus nucleus. This raises the question whether spatial noise correlations within the medial rectus nucleus might influence our conclusions that OMN spiking noise cannot account for the MSD of fixational drift. This is highly unlikely for the following reasons:

- (i) Ref. 53 shows that antidromic activation in the medial rectus does not produce any synaptic responses within the nucleus. This suggests that the collaterals in the medial rectus nucleus do not form recurrent synaptic connections within the nucleus, which could in principle drive noise correlations. Based on this observation, as well as other studies, it was possible to conclude that there are no recurrent synaptic connections from ocular motoneurons to other ocular motoneurons (see Ref. 53 in the revised submission).
- (ii) Stimulation studies indicate that there is no significant *temporal* integration within the abducens and medial rectus nuclei beyond the characteristic response time of ~180 ms which we have already taken into account in our model (Ref. 54 and 55). Therefore, even if there is some level of spatial noise correlation between OMNs which is intrinsically generated in these nuclei, and which we have incorrectly neglected, the temporal aspects of the response are correctly taken into account in our model. The implication would be to multiplicatively amplify the variance of the motion, but not to affect the logarithmic slope of the MSD curve. In other words, such correlations would shift the MSD curve vertically in the logarithmic plot, and this would not produce an MSD curve that matches the actual eye motion at long time scales. This is demonstrated by the dashed line in Fig. 3b (previously Fig. 3), which we now discuss also in this context.

We now discuss these points in lines 301-311 of the revised manuscript.

Line 122 Does MSD refer only to horizontal displacement?

Yes. We now point out more explicitly around this line, and elsewhere in the manuscript, that we analyze the horizontal eye motion.

Fig 2e More explanation of the axes is needed – is this a log scale?

(Note, use of two differently typeset epsilons to indicate distinct quantities makes it hard to refer to)

Thanks for this comment. We realize now that Fig. 2e was difficult to digest on a first read, because of the relatively unconventional choice of axes: for each symbol (corresponding to one OMN) the horizontal position represents an estimate of χ , and the vertical position represents our estimation error of this quantity (the size of the confidence interval). Therefore, we added a new panel (now Fig. 2e) which shows that data from one monkey in a simpler form, which is

easier to understand on a first read. For the original Fig. 2e (now Fig. 2f) we added clarifying text both in the main text and in the figure caption, and relabeled the y-axis.

Line 315: cite evidence that OMNs have negligible noise correlations to support this assumption

Note that we refer here only to noise which is intrinsic to the OMNs, as opposed to correlations arising from upstream inputs. We based our assumption on the lack of collaterals within the abducens nucleus, as well as stimulation studies (see above). This is now discussed in a new paragraph in the Discussion section.

Line 374: Were no primate data available to estimate the tuning functions? What is the rationale for the range chosen?

Thanks for this comment. We found data on the tuning curves of neurons which are suspected to compose the oculomotor integrator in monkeys (McFarland, Fuchs, 1992). Specifically, this work identifies a relation between the eye position sensitivity and eye threshold. We revised our simulations, used this relationship to sample the parameters of the simulated oculomotor integrator neurons, and updated our fit to the data (and the Methods section) accordingly.

The main difference with respect to the previous submission is as follows: previously the averaged eye position sensitivity parameter was $k \sim 7 \text{ spk} \cdot \text{deg}^{-1} \cdot \text{s}^{-1}$, but the averaged value according to Fuchs paper is $k \sim 3 \text{ spk} \cdot \text{deg}^{-1} \cdot \text{s}^{-1}$. This difference has an immediate implication on the amplitude of the resulting MSD curve (according to Burak, Fiete, 2012), since the MSD is proportional to $\sim CV^2 \cdot N^{-1} \cdot \tau_s^{-2} \cdot k^{-2} \cdot r^{-1}$, where CV is the typical coefficient of variation of inter-spike intervals, N is the number of neurons composing the network, τ_s is the synaptic time constant, k is the typical eye position sensitivity, and r is the typical firing rate. Therefore, in order to compensate for the reduction in the typical eye position sensitivity we adjusted two parameters, one is the synaptic time constant, τ_s , that was changed from 10ms to 20 ms, and second, the number of neurons in the oculomotor integrator network, which was changed from 20k to 30k. Note that other combinations of τ_s and N could have been used, since they contribute together to the MSD according to the mathematical relation mentioned above, and we now elaborate on this in the manuscript (Lines 242-254).

Line 425 unneeded comma "where the ISI, is measured"

Thanks, fixed.

EFig 9: How sensitive is the model to the distribution of integrator tuning functions? Is there any data in the primate literature to support the choice made?

Following up on our reply to the comment above on line 374: Ultimately, what matters for the magnitude of the motion is the diffusivity along the attractor, which depends on a combination of many parameters: the individual tuning curves, as well as other parameters such as the synaptic time constant, the number of neurons, etc. Reference 32 provides a detailed analysis of how the diffusivity in the oculomotor integrator depends on the individual tuning curves of all the neurons that participate in the line attractor dynamics. Roughly speaking, the diffusivity is proportional to an average over the population of $k^{-2} \cdot r^{-1}$ where k is the eye position sensitivity and r is the firing rate at the fixation point. Since we only have approximate estimates for these quantities, our goal in the detailed simulation of the oculomotor integrator is not to precisely predict the diffusivity, but to show that when using reasonable estimates, the diffusion in the oculomotor integrator drives eye motion with variance that matches the actual variance of fixational drift. This is now explained explicitly in the text.

Revision Figure 1. Correlations of OMNs parameters, k –position sensitivity, r –velocity sensitivity, E_T –position threshold, rate–the averaged firing rate during fixation, CV–the averaged CV during fixation, and their correlation with the measured eye position and χ (left column in

each panel, headed “data”). For comparison we present similar analysis over simulated data (right column in each panel, headed “simulation”). Simulations include 100 trials at center gaze and 30 trials at 5 deg and 10 deg offset fixations, in similarity to the empirical data. For the presented OMNs in simulations we used similar parameters as we measured from the data. In all of the panels the Spearman correlation coefficient is presented at the title, along its p-value.

Revision Figure 2. The correlation coefficient of each OMNs predicted eye position with the measured eye position vs the OMN peak kernel value. The majority of the OMNs has kernel peak value in the range of 1.5-3 deg/spk. There is no significant correlation between these variables, Spearman correlation coefficient of -0.16 with p-value=0.24.

REVIEWER COMMENTS

Reviewer #1 (Remarks to the Author):

The manuscript is improved over the previous version, but there are still issues that need to be addressed by the authors:

- line 39: but people have actually related the representation of ocular target goals in the superior colliculus to smooth drifts in eye position. e.g. Hafed et al., J. Neurosci., 2008; Hafed & Krauzlis, J. Neurosci., 2008; Goffart et al., J. Neurosci., 2012; Chen et al., Curr Biol., 2019. This is also reviewed in detail in Hafed et al., J. Neurophysiol., 2021 (e.g. their Fig. 8). See also later comment about the discussion.

- line 54: say “in more central neural circuitry” not “in central neural circuitry”

- what does shuffling mean in Fig. 2d? This is not explained in either the text describing the figure or the figure legend itself. This needs to be explained, and the statement that this histogram implies “noisy distribution” needs to be clarified.

- line 99-103: I’m still very confused here. The authors refer to R^2 and simultaneously refer to Fig. 2d, but Fig. 2d shows R and not R^2 . Then, they say that the average of R^2 is larger than the square of the average R , but I don’t see this anywhere. I don’t see any of the numbers that show that the average of R^2 is larger than the square of the average of R .

- line 106: it is not clear what is meant by OMN’s contributing “independently” to the eye motion. Later text helps a bit, but this needs to be clarified. Also, it seems that such a counter-argument seems to be unnecessary (straw man). The OMN’s are the actuator of the muscle, so why would they not reflect the muscle motions?

- line 291: this sentence needs to be much better qualified by saying something like “...correlated with neural activity in the oculomotor control circuitry of the brainstem”. The reason is that correlations to neural activity in other brain areas have indeed been shown before (e.g. the superior colliculus stuff mentioned above, but also the primary visual cortex stuff of Kagan and Snodderly, etc).

- line 305: I do not understand this sentence. Does it mean to say that the abducens neurons do not project to the abducens nucleus locally?

- line 325: one important qualification to this statement is the Malevich et al., eLife, 2020 study that you cited earlier. The visual influence on drifts in that study is quite fast.

Reviewer #2 (Remarks to the Author):

The revised manuscript nicely clarifies and adds detail to the prior descriptions. The paper as written reads very well and my previous points are addressed in full. Congratulations to the authors on their excellent research!

Reviewer #3 (Remarks to the Author):

All of my major concerns have been resolved. I only have a few minor comments and suggestions.

Line 36ff: "However, fixational drift is observed even in complete darkness [26]. Therefore, it is unlikely that the main drive for the motion originates in visual feedback mechanisms."

I understand the point, but this seems a bit simplistic as a dismissal of a visual origin. For example, substantial noise in feedback pathways in darkness might contribute to drift. Perhaps an additional sentence might flush this line of argument more convincingly.

Figs 1D and 2D Consider adding the population mean(s) to the panels.

Line 133ff: "Thus, the covariance between the predicted motion and the measured eye motion extracts, from the overall variability in the measured eye motion, the variance that originates in the central source (see also Methods)."

You might comment on Rodriguez-Falces, Negro, Farina J Neurophys 2017 here since they reach a different conclusion about assessing the fraction of shared variability in motor units.

Line 180: "Similar saturation is expected to arise from any form of temporally uncorrelated noise which is fed into the muscle dynamics, at time lags exceeding the characteristic time scale of the muscle response (~ 180 ms)."

A citation, or an expanded argument, would be helpful here.

Line 240 "...which was measured..."  were

We thank all the reviewers for providing additional comments on the manuscript. We addressed all of these comments in our revision.

Reviewer #1 (Remarks to the Author):

The manuscript is improved over the previous version, but there are still issues that need to be addressed by the authors:

We addressed the remaining comments (see below).

- line 39: but people have actually related the representation of ocular target goals in the superior colliculus to smooth drifts in eye position. e.g. Hafed et al., J. Neurosci., 2008; Hafed & Krauzlis, J. Neurosci., 2008; Goffart et al., J. Neurosci., 2012; Chen et al., Curr Biol., 2019. This is also reviewed in detail in Hafed et al., J. Neurophysiol., 2021 (e.g. their Fig. 8). See also later comment about the discussion.

Thanks for this comment. We changed the first sentence of line 39, from:

“So far, no evidence has directly linked fixational drift to neural activity”,

to:

“So far, direct evidence for the control of fixational eye drifts by neural activity has been lacking”,

and cited the J. Neurophysiol. Review by Hafed et al. The references mentioned above provide compelling arguments for an involvement of the SC in fixational drifts. We now address this point in the discussion section (lines 336-342 in the revised manuscript).

- line 54: say “in more central neural circuitry” not “in central neural circuitry”

Done.

- what does shuffling mean in Fig. 2d? This is not explained in either the text describing the figure or the figure legend itself. This needs to be explained, and the statement that this histogram implies “noisy distribution” needs to be clarified.

We clarified this in the Methods section (lines 428-433 in the revised manuscript) and pointed to the Methods section in the caption of Figure 2.

- line 99-103: I’m still very confused here. The authors refer to R^2 and simultaneously refer to Fig. 2d, but Fig. 2d shows R and not R squared. Then, they say that the average of R squared is larger than the square of the average R , but I don’t see this anywhere. I don’t see any of the numbers that show that the average of R squared is larger than the square of the average of R .

As pointed out in the reviewer’s comment, Fig. 2d shows a histogram of the measurements of R from different neurons. The measurements themselves are shown in SI Fig. 1a-b. It is

possible to generate an estimate of $\langle R \rangle$ using these measurements, as we did in the manuscript. Alternatively, these measurements can be squared and averaged across cells to generate an estimate of $\langle R^2 \rangle$. This estimate ends up being around 0.1. We were concerned that large values of R which originate from measurement errors might result in an overestimate of $\langle R^2 \rangle$. Therefore, we preferred to be more conservative and use $\langle R \rangle$ to obtain a lower bound on $\langle R^2 \rangle$.

The fact that $\langle R^2 \rangle \geq \langle R \rangle^2$ is a mathematical identity, which applies to any distribution, due to the fact that the variance (which is the difference between these two quantities) is always positive. For this reason, it is possible to use the empirical average of R in order to bound from below the average of R^2 . These points are explained in the Supplementary Notes.

We now refer to SI Figs. 1a-b (which include the correlation coefficients from individual neurons) in addition to Fig. 2, in lines 104-106 in the revised manuscript, and we added a few more sentences to explain more clearly that we extracted $\langle R \rangle$ from the distribution shown in these figures (lines 105-106).

- line 106: it is not clear what is meant by OMN's contributing "independently" to the eye motion. Later text helps a bit, but this needs to be clarified. Also, it seems that such a counter-argument seems to be unnecessary (straw man). The OMN's are the actuator of the muscle, so why would they not reflect the muscle motions?

Thanks for the question. We don't think that the scenario that we discuss is a straw-man, but we see how the phrasing could have been insufficiently clear. The scenario that we consider is one where spiking noise in the OMNs (as opposed to an upstream drive feeding into the OMNs) generates the main drive for the motion. This a-priori might seem possible because fixational drift is quite tiny, and the readout from individual OMNs is much more variable than this motion. To clarify what we meant we replaced:

"For comparison, consider a scenario in which OMNs contribute independently to the eye motion."

By:

"For comparison, consider a scenario in which the upstream drive to OMNs is completely static during fixational drift, and the motion is driven by the intrinsic variability of the OMNs, which is independent in the different neurons."

- line 291: this sentence needs to be much better qualified by saying something like "...correlated with neural activity in the oculomotor control circuitry of the brainstem". The reason is that correlations to neural activity in other brain areas have indeed been shown before (e.g. the superior colliculus stuff mentioned above, but also the primary visual cortex stuff of Kagan and Snodderly, etc).

Thanks for the clear suggestion of a way to fix this, which addresses also the issues raised in the first comment. We agree and have modified the text accordingly (now in lines 296 of the revised manuscript).

- line 305: I do not understand this sentence. Does it mean to say that the abducens neurons do not project to the abducens nucleus locally?

The implication of the lack of collaterals is that abducens neurons are unlikely to have targets other than the muscle. In the medial rectus nucleus collaterals do exist, but they do not terminate within the nucleus. The implication is the OMNs in the medial rectus nucleus do not project to the nucleus locally. We clarified this in the text (lines 311-312 in the revised manuscript).

- line 325: one important qualification to this statement is the Malevich et al., eLife, 2020 study that you cited earlier. The visual influence on drifts in that study is quite fast.

Thanks for pointing this out. Of course, 'fast' and 'slow' are a matter of terminology. We assumed a 70ms delay in our model, which is in agreement with the delays reported on by Malevich et al (ranging from 60ms to 80ms in different monkeys). We now mention this range explicitly, and have replaced 'large synaptic delays' by 'synaptic delays' because the classification of delays in the range of 60-80ms as large is subjective.

Reviewer #2 (Remarks to the Author):

The revised manuscript nicely clarifies and adds detail to the prior descriptions. The paper as written reads very well and my previous points are addressed in full. Congratulations to the authors on their excellent research!

Thanks for this positive assessment, and for the helpful suggestions in the previous report.

Reviewer #3 (Remarks to the Author):

All of my major concerns have been resolved. I only have a few minor comments and suggestions.

Thanks for the comments and suggestions. We have addressed these comments in the revised manuscript: please see our detailed responses below.

*Line 36ff: "However, fixational drift is observed even in complete darkness [26]. Therefore, it is unlikely that the main drive for the motion originates in visual feedback mechanisms."
I understand the point, but this seems a bit simplistic as a dismissal of a visual origin. For example, substantial noise in feedback pathways in darkness might contribute to drift. Perhaps an additional sentence might flush this line of argument more convincingly.*

We agree that noise originating in neural circuits that are involved in visual feedback mechanisms might contribute to fixational drift, and we didn't intend to rule out this possibility (see also lines 324-326 in the Discussion). We changed the text (lines 36-39 in the revised manuscript) as follows:

“However, an active visuomotor response to a stimulus is not required to elicit fixational drift, since it is observed even in complete darkness. The stochastic nature of the motion, both in the presence and in the absence of visual stimuli, suggests that it is primarily driven by noise which may arise in various stages along the oculomotor pathway.”

Figs 1D and 2D Consider adding the population mean(s) to the panels.

We added the means.

Line 133ff: “Thus, the covariance between the predicted motion and the measured eye motion extracts, from the overall variability in the measured eye motion, the variance that originates in the central source (see also Methods).”

You might comment on Rodriguez-Falces, Negro, Farina J Neurophys 2017 here since they reach a different conclusion about assessing the fraction of shared variability in motor units.

Thanks for the comment. The reference above explores the possibility of identifying a common drive (to motoneurons that innervate the biceps brachii muscle) based on correlations in the spiking activity of two motoneurons, and concludes that it is very difficult to do so. Since we did not have simultaneous recordings of multiple OMNs, these difficulties are not directly relevant to our approach. Note also that the manuscript of Rodriguez-Falces et al does not attempt to identify a common drive based on the covariance of a single OMN readout and the motor output, as in our work.

Line 180: “Similar saturation is expected to arise from any form of temporally uncorrelated noise which is fed into the muscle dynamics, at time lags exceeding the characteristic time scale of the muscle response (~ 180 ms).”

A citation, or an expanded argument, would be helpful here.

We added an explanation of this statement in the Methods section (lines 486-495), and referred to it in line 187.

Line 240 “...which was measured...”  were

Corrected.

REVIEWERS' COMMENTS

Reviewer #1 (Remarks to the Author):

The authors have addressed all of the previous reviewer comments.

Reviewer #3 (Remarks to the Author):

My concerns have been resolved